# DiffuCoder: Understanding and Improving Masked Diffusion Models for Code Generation

**Shansan Gong**[1‡2], **Ruixiang Zhang**[1], **Huangjie Zheng**[1], **Jiatao Gu**[1], **Navdeep Jaitly**[1†],
**Lingpeng Kong**[2†], **Yizhe Zhang**[1]

[1]Apple    [2]The University of Hong Kong [†]Core advising
sansa933@connect.hku.hk;lpk@cs.hku.hk;yizhe_zhang@apple.com

[‡] Work done during the internship at Apple

## Abstract

Diffusion large language models (dLLMs) are compelling alternatives to autoregressive (AR) models because their denoising models operate over the entire sequence. The global planning and iterative refinement features of dLLMs are particularly useful for code generation. However, current training and inference mechanisms for dLLMs in coding are still under-explored. To demystify the decoding behavior of dLLMs and unlock their potential for coding, we systematically investigate their denoising processes and reinforcement learning (RL) methods. We train a 7B dLLM, **DiffuCoder**, on 130B tokens of code. Using this model as a testbed, we analyze its decoding behavior, revealing how it differs from that of AR models: (1) dLLMs can decide how causal their generation should be without relying on semi-AR decoding, and (2) increasing the sampling temperature diversifies not only token choices but also their generation order. This diversity creates a rich search space for RL rollouts. For RL training, to reduce the variance of token log-likelihood estimates and maintain training efficiency, we propose **coupled-GRPO**, a novel sampling scheme that constructs complementary mask noise for completions used in training. In our experiments, coupled-GRPO significantly improves DiffuCoder's performance on code generation benchmarks (+4.4% on EvalPlus) and reduces reliance on AR bias during decoding. Our work provides deeper insight into the machinery of dLLM generation and offers an effective, diffusion-native RL training framework. https://github.com/apple/ml-diffucoder

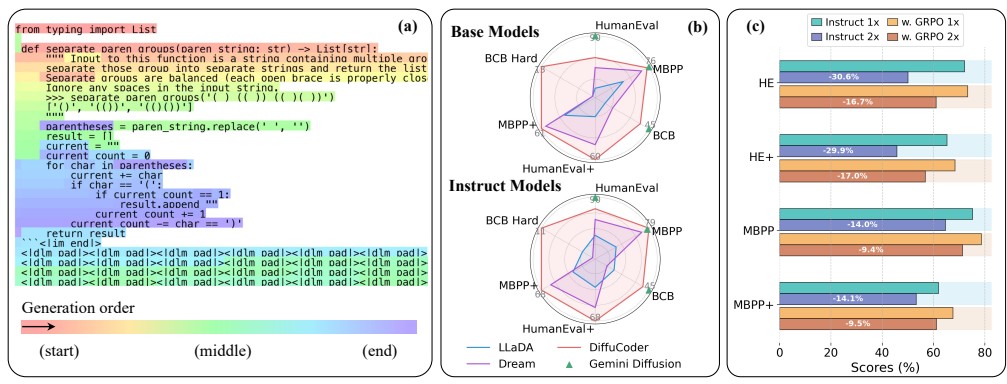

Figure 1: (a) A real example of DiffuCoder-Instruct's decoding process with sampling temperature 1.2. (b) Results on coding benchmarks. (c) When decoding steps are halved, DiffuCoder-Instruct trained with coupled-GRPO experiences a smaller performance drop, compared to Instruct itself.

## 1 Introduction

Large language models (LLMs) have shown remarkable capabilities across tasks, from dialogue to code generation (Touvron et al., 2023; OpenAI, 2023). While these successes are built predominantly

on the autoregressive (AR) paradigm, masked diffusion models (MDMs) have recently emerged as a compelling alternative (Zheng et al., 2024; Shi et al., 2024; Sahoo et al., 2024), and are further scaled to diffusion LLMs (dLLMs) like LLaDA (Nie et al., 2024) and Dream (Ye et al., 2025), which achieve performance on par with similarly sized AR LLMs. Rather than generating left-to-right, MDMs iteratively refine the entire sequence in parallel, which allows for global planning of content (Ye et al., 2024a; Zhang et al., 2023) and also improves flexibility during decoding (Kim et al., 2025).

Intuitively, code generation aligns well with the dLLM paradigm, as writing code often involves non-sequential back and forth refinement (Xie et al., 2025). Recent commercial-scale dLLMs (Inception Labs et al., 2025; DeepMind, 2025) show that a diffusion-based code generator can rival top AR code models. However, it remains unclear how open-source dLLMs perform on coding tasks, as their training and inference mechanisms are not yet fully interpreted. Existing post-training efforts for dLLMs, such as LLaDA1.5 (Zhu et al., 2025) with DPO (Rafailov et al., 2023) training, and d1 (Zhao et al., 2025), MMaDA (Yang et al., 2025) with GRPO (Shao et al., 2024) training, either show marginal gains or rely heavily on semi-AR decoding (*i.e.*, block decoding with a relatively small block size; see Arriola et al., 2025) which deviates from the global planning nature of diffusion. To address these limitations, we first gain insight into decoding behaviors of dLLMs and then establish a diffusion-native reinforcement learning (RL) methodology.

Our investigation is grounded in the analysis of **DiffuCoder**, a 7B-scale MDM specialized for code generation (§3), trained on 130B effective tokens (Huang et al., 2024). The model's performance is competitive with that of AR coders, providing a strong testbed for understanding the behaviors of dLLMs and for developing diffusion-native post-training approaches.

To leverage the benefits of non-autoregressiveness in dLLMs, it is important to understand how non-autoregressive the behavior of current dLLMs actually is. To this end, we introduce local and global *autoregressive-ness* (AR-ness) metrics (§4.1) to measure how closely their generation follows a left-to-right pattern. We show that DiffuCoder can automatically decide how non-autoregressive it needs to be during decoding (§4.2). When the sampling temperature is increased from the default 0.2 to 1.2, DiffuCoder becomes more flexible in its token generation order, freeing itself from strict left-to-right constraints, as Figure 1(a) shows. Unlike AR models, which primarily diversify token choices at higher temperatures, dLLMs additionally diversify the position of the generated token. With this increased diversity, DiffuCoder achieves higher pass@10 accuracy by changing the sampling temperature from 0.2 to 1.2 in our experiments (§4.3). The gain in pass@10 indicates the potential capacity of DiffuCoder, suggesting it can benefit from effective RL training to "elicit out" the most successful rollout samples.

Following this, we tailor GRPO (Shao et al., 2024) for dLLMs. Our design focuses on reducing variance while maintaining efficiency in Monte Carlo estimations of token likelihoods. Specifically, we propose **coupled-GRPO**, which employs a novel coupled-sampling scheme. In detail, it adds paired complementary mask noise to the completion sequences generated by the model at a temperature of 1.2. Unlike previous approaches (Zhao et al., 2025), our method does not rely on semi-AR decoding and it further improves the instruct-tuned model. After coupled-GRPO training, the model exhibits a stronger non-AR generation pattern, as inferred from Figure 1(c).

In summary, our contributions are as follows.

- We introduce a 7B dLLM for code generation, providing a foundation for developing diffusion-native training methods (§3).

- We introduce local and global *AR-ness* metrics to demystify the decoding patterns of dLLMs and track how AR-ness evolves across different training stages (§4.2). Our analysis reveals that higher sampling temperatures encourage more parallel, non-AR generation (§4.3).

- We design coupled-GRPO, an RL algorithm for dLLMs that avoids semi-AR decoding by using a novel coupled-sampling scheme for efficient and accurate policy gradient estimation (§5). We theoretically prove the variance reduction of coupled-GRPO using antithetic variates.

- Experimentally, coupled-GRPO significantly improves DiffuCoder's performance, boosting its EvalPlus score by 4.4% with training on only 21K samples and demonstrating the effectiveness of RL aligned with diffusion principles.

## 2 PRELIMINARIES AND NOTATIONS

### 2.1 MASK DIFFUSION MODELS

In diffusion models (Ho et al., 2020; Song et al., 2021), the forward process $q(\boldsymbol{x}_{1:T}|\boldsymbol{x}_0) = \prod_{t=1}^{T} q(\boldsymbol{x}_t|\boldsymbol{x}_{t-1})$ gradually corrupts data $\boldsymbol{x}_0 \sim p_{data}(\boldsymbol{x}_0)$ into noisy variables $\boldsymbol{x}_{1:T}$. The backward process models the joint probability as $p_\theta(\boldsymbol{x}_{0:T}) = p_\theta(\boldsymbol{x}_T)\prod_{t=1}^{T} p_\theta(\boldsymbol{x}_{t-1}|\boldsymbol{x}_t)$, denoising $\boldsymbol{x}_t$ to reconstruct $\boldsymbol{x}_0$. Discrete diffusion models (Hoogeboom et al., 2021; Zheng et al., 2024) define the forward process with a categorical distribution $q(\boldsymbol{x}_t|\boldsymbol{x}_{t-1}) = \text{Cat}(\boldsymbol{x}_t; \boldsymbol{Q}_t^\top \boldsymbol{x}_{t-1})$, where $\boldsymbol{x}_t \in \{0,1\}^K$ is a one-hot vector with vocabulary size $K$, and $\boldsymbol{Q}_t \in [0,1]^{K \times K}$ is the transition matrix where $[\boldsymbol{Q}_t]_{ij}$ represents the probability of transition from state $i$ to $j$. For absorbing discrete diffusion (Austin et al., 2021a), $\boldsymbol{Q}_t = (1 - \beta_t)I + \beta_t \mathbf{1}\boldsymbol{m}^\top$, where $\mathbf{1}$ is an all-one vector of size $K$ and $\boldsymbol{m}$ is the one-hot encoding of a special [MASK] token. The parameters $\theta$ are learned by minimizing the negative log-likelihood of $\boldsymbol{x}_0$ through the evidence lower bound (ELBO). For continuous time modeling (Shi et al., 2024; Sahoo et al., 2024), the discrete timesteps $t = \{1 \ldots T\}$ are scaled to a mask ratio within $[0,1]$, yielding the final ELBO at a sampled time $t$ as a weighted cross-entropy loss (Ou et al., 2024):

$$\mathcal{L}_t^{1:N} = \frac{1}{t}\mathbb{E}_{q(\boldsymbol{x}_t|\boldsymbol{x}_0)}\left[-\sum_{n=1}^{N}\delta_{\boldsymbol{x}_t^n,\boldsymbol{m}}(\boldsymbol{x}_0^n)^\top \log f_\theta(\boldsymbol{x}_t^{1:N})_n\right], \tag{1}$$

where $\delta_{\boldsymbol{x}_t^n,\boldsymbol{m}}$ is the indicator function, and $f_\theta(\boldsymbol{x}_t^{1:N})_n$ represents the logits for the $n$-th token given the $N$-length input sequence. Please refer to detailed related work in §6.

### 2.2 MARKOV DECISION PROCESS

DDPO (Black et al., 2024) and DPPO (Ren et al., 2025) reinterpret the denoising diffusion process as a Markov Decision Process (MDP). An MDP is a tuple $\mathcal{M}_{env} = (\mathcal{S}, \mathcal{A}, P_0, P, R)$ with a state space $\mathcal{S}$, an action space $\mathcal{A}$, an initial state distribution $P_0$, transition probabilities $P$, and a reward function $R$. The probability of transitioning to state $s_{t+1}$ is $P(s_{t+1} \mid s_t, a_t)$ after taking an action $a_t \sim \pi_\theta(a_t \mid s_t)$ and receiving a reward $R(s_t, a_t)$. The goal is to maximize the expected return $\mathcal{J}(\pi_\theta) = \mathbb{E}_{\pi_\theta}\left[\sum_{t=0}^{T}\gamma(t)R(s_t, a_t)\right]$ by using the policy gradient method (Williams, 1992):

$$\nabla_\theta \mathcal{J}(\pi_\theta) = \mathbb{E}_{\pi_\theta}\Big[\sum_{t=0}^{T}\nabla_\theta \log \pi_\theta(a_t|s_t)\,r_t(s_t, a_t)\Big]; \quad r_t(s_t, a_t) = \sum_{\tau \geq t}\gamma(\tau)R(s_\tau, a_\tau). \tag{2}$$

In a masked diffusion model for conditional generation, we set the state to $s_t = (c, t, \boldsymbol{x}_t)$ (where $c$ is the condition) and the action to $a_t = \boldsymbol{x}_{t-1}$, so that $\pi_\theta(a_t|s_t) = p_\theta(\boldsymbol{x}_{t-1}|\boldsymbol{x}_t, c)$. We sample $c \in \mathcal{C}$ and trajectories of $\boldsymbol{x}_t$. The reward is defined as $R(s_0, a_0) = r(\boldsymbol{x}_0, c)$ at the final denoising step because a fine-grained progressive reward is usually hard to quantify, especially in intermediate diffusion steps. Under this setting, the policy gradient becomes: $\nabla_\theta \mathcal{J} = \mathbb{E}\big[\sum_{t=0}^{T}\nabla_\theta \log p_\theta(\boldsymbol{x}_{t-1}|\boldsymbol{x}_t, c)\,r(\boldsymbol{x}_0, c)\big]$.

### 2.3 GROUP RELATIVE POLICY OPTIMIZATION

GRPO (Shao et al., 2024) simplifies PPO (Schulman et al., 2017). It samples a group of outputs $\{o_i\}_{i=1}^{G}$ from the old policy $\pi_{\theta_{old}}$ under a given condition $c$, estimates the value baseline by averaging rewards within the group, and defines the relative advantage for each output as $A_i = r(o_i) - \frac{1}{G}\sum_{j=1}^{G}r(o_j)$. For token $1 \leq k \leq |o_i|$, we denote $\rho_i^k = \frac{\pi_\theta(o_i^k|c, o_i^{<k})}{\pi_{old}(o_i^k|c, o_i^{<k})}$ as the token-level importance ratio. The GRPO loss applies a PPO-style clipping to $\rho_i^k$, incorporating a KL-penalty to keep $\pi_\theta$ close to a reference policy $\pi_{ref}$, and maximizes the following surrogate objective:

$$\mathcal{J}_{\text{GRPO}}(\theta) = \mathbb{E}_{o_i \sim \pi_{\theta_{old}}}\Big[\sum_{i=1}^{G}\sum_{k=1}^{|o_i|}\min\big(\rho_i^k A_i, \text{clip}(\rho_i^k, 1-\varepsilon, 1+\varepsilon)A_i\big) - \beta\,D_{\text{KL}}\big(\pi_\theta\|\pi_{ref}\big)\Big]. \tag{3}$$

By estimating the group mean via Monte Carlo estimation, GRPO avoids training a separate value function while fitting neatly into the MDP framework. As the diffusion process can be viewed as an MDP, the GRPO loss can be applied to MDMs by combining Eq. (2) and Eq. (3).

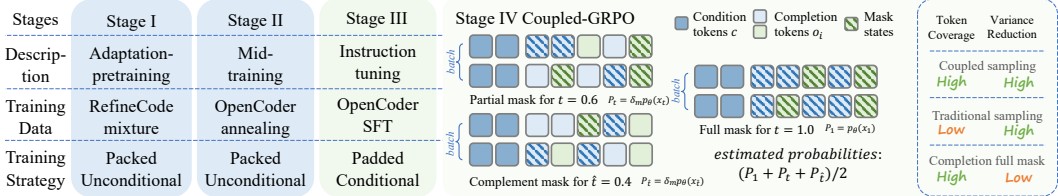

Figure 2: Pipeline of DiffuCoder training stages and an illustration of the coupled-GRPO algorithm. We sample complementary mask matrices for the same batch, so the coupling probability matrices can be merged into one full matrix. Coupled sampling reduces probability estimation variance while maintaining full token coverage, where each token is sampled exactly the same number of times.

## 3 DIFFUCODER

To build a strong base model for RL training, we follow common practices in LLM training and train our DiffuCoder model on a large-scale corpus (Lozhkov et al., 2024) with multiple training stages; the pipeline is illustrated in Figure 2. We first conduct adaptation pre-training similar to the process followed by Dream (Ye et al., 2025). Mid-training (Wang et al., 2025) connects the pre-training and post-training stages, plays the role of an annealing phase as in OpenCoder (Huang et al., 2024), and has proven effective. This is followed by an instruction tuning stage to enhance the model's capability of following instructions. Finally, for post-training, we employ a novel **coupled-GRPO** method (introduced in §5) to further enhance the model's pass@1 coding capabilities.

**Training**  We adapt our model from Qwen-2.5-Coder (Hui et al., 2024) as the base model to perform continual pre-training using the adaptation approach from Gong et al. (2025). During this pre-training, we use a 400B-token code pre-training corpus from RefineCode (Huang et al., 2024) and Stackv2 (Lozhkov et al., 2024). We adopt the code-to-text ratio suggested in Qwen-2.5-Coder and OpenCoder (Huang et al., 2024). We use 16B tokens of annealing code data during mid-training and 436K SFT samples during instruction tuning, both from OpenCoder (Huang et al., 2024). For RL training, we select 21K hard samples from Acecoder-87K (Zeng et al., 2025) with verifiable test cases. We build our post-training method upon the Open-R1[1] codebase. All experiments are conducted on 8 to 10 nodes, each with 8 H100 GPUs. We observed that training with 700B tokens in Stage 1 led to worse performance than using only 65B tokens on the downstream validation sets, probably because the continual pre-training is sensitive to the data quality. Therefore, we perform early stopping for Stage 1, training on 65B tokens. In Stage 2, we train for 4 epochs, totaling 65B tokens with repeats, as the mid-training data is less noisy. More details are listed in Appx. B.1.

**Evaluation**  Our evaluation environments are built on three code benchmarks and their variants: HumanEval (Chen et al., 2021), MBPP (Austin et al., 2021b), EvalPlus (HumanEval+ and MBPP+; Liu et al. 2023), and BigCodeBench (Zhuo et al., 2024) with full and hard subsets in completion (C) and instruction (I) query types. These benchmarks, with Python as the coding language, provide a diverse set of coding tasks for assessing code correctness and quality.

**Performance**  We compare our models with AR code LLMs, including Qwen2.5-Coder-7B (Hui et al., 2024), OpenCoder-8B (Huang et al., 2024); general dLLMs, such as Dream-7B (Ye et al., 2025) and LLaDA-8B (Zhu et al., 2025); and commercial models, such as GPT-4o[2], Mercury[3], and Gemini Diffusion[4]. As shown in Table 1, DiffuCoder, after being continually trained on 130B code tokens (Stages 1 and 2), achieves performance on par with Qwen2.5-Coder and OpenCoder. However, all dLLMs show only marginal improvement over their base models after instruction tuning, especially when compared to Qwen2.5-Coder+SFT, which achieves large improvements from being instruct-tuned on the same data.

---

[1] https://github.com/huggingface/open-r1
[2] https://openai.com/index/gpt-4o-system-card/
[3] https://chat.inceptionlabs.ai/
[4] https://deepmind.google/models/gemini-diffusion/

Table 1: Benchmark coding capacities of LLMs and dLLMs in 7/8B scale. Different shaded colors indicates different generation paradigms (pink for AR, yellow for diffusion). * denotes that the results are collected from public reports instead of evaluating by ourselves. We compute EvalPlus as the average of HE+ and MBPP+ and show the absolute score change (±) of each instruct model relative to its base. We bold scores when our model outperforms at least one LLMs.

| Model | HumanEval | | MBPP | | EvalPlus | BigCodeBench (C) | | Avg. |
|---|---|---|---|---|---|---|---|---|
| | - | Plus | - | Plus | | Full | Hard | |
| **Base Models** | | | | | | | | |
| Qwen2.5-Coder | 61.6 | 51.8 | 75.9 | 61.4 | 56.6 | 46.1 | 16.2 | 52.2 |
| OpenCoder* | 66.5 | 63.4 | 79.9 | 70.4 | 66.9 | 40.5 | 9.5 | 55.0 |
| LLaDA | 35.4 | 30.5 | 50.1 | 42.1 | 36.3 | 18.9 | 4.1 | 30.2 |
| Dream | 56.7 | 50.0 | 68.7 | 57.4 | 53.7 | 23.6 | 4.1 | 43.4 |
| DiffuCoder | **67.1** | **60.4** | 74.2 | 60.9 | **60.6** | 40.2 | **12.8** | **52.6** |
| **Instruct Models** | | | | | | | | |
| Qwen2.5-Coder-Instruct | 90.2 | 85.4 | 83.9 | 72.0 | 78.7 | 50.7 | 21.6 | 67.3 |
| Qwen2.5-Coder+SFT | 82.9$_{+21.3}$ | 75.6$_{+23.8}$ | 80.1$_{+4.2}$ | 66.1$_{+4.7}$ | 70.9$_{+14.3}$ | 46.9$_{+0.8}$ | 16.2$_{-0.0}$ | 61.3$_{+9.1}$ |
| OpenCoder-Instruct* | 83.5$_{+17.0}$ | 78.7$_{+15.3}$ | 79.1$_{-0.8}$ | 69.0$_{-1.4}$ | 73.9$_{+7.0}$ | 40.3$_{-0.2}$ | 16.9$_{+7.4}$ | 61.3$_{+6.3}$ |
| LLaDA-Instruct | 35.4$_{-0.0}$ | 31.7$_{+1.2}$ | 31.5$_{-18.6}$ | 28.6$_{-13.5}$ | 30.2$_{-6.1}$ | 16.5$_{-2.4}$ | 2.7$_{-1.4}$ | 24.4$_{-5.8}$ |
| Dream-Instruct | 57.9$_{+1.2}$ | 53.7$_{+3.7}$ | 68.3$_{-0.4}$ | 56.1$_{-1.3}$ | 54.9$_{+1.2}$ | 10.6$_{-13.0}$ | 0.7$_{-3.4}$ | 41.2$_{-2.2}$ |
| DiffuCoder-Instruct | 72.0$_{+4.9}$ | 65.2$_{+4.8}$ | 75.1$_{+0.9}$ | 61.9$_{+1.0}$ | 63.6$_{+3.0}$ | 35.7$_{-4.5}$ | 12.2$_{-0.6}$ | 53.7$_{+1.1}$ |
| + coupled-GRPO | 73.2$_{+6.1}$ | 68.3$_{+7.9}$ | **78.6**$_{+4.4}$ | 67.5$_{+6.6}$ | 67.9$_{+7.3}$ | **40.4**$_{+0.2}$ | 10.8$_{-2.0}$ | 56.5$_{+3.9}$ |
| **Commercial Models** | | | | | | | | |
| GPT 4o* | 90.2 | – | 82.2 | – | 82.4 | 49.9 | – | – |
| Mercury* | 90.0 | – | 77.1 | – | 80.4 | 45.5 | – | – |
| Gemini Diffusion* | 89.6 | – | 76.0 | – | – | 45.4 | – | – |

This improvement gap between AR and dLLMs at the instruction-tuning stage motivates us to explore RL-based post-training methods (§5). Previous RL approaches for diffusion models (Zhao et al., 2025; Yang et al., 2025) rely heavily on semi-AR decoding, which deviates from diffusion's global nature. To design RL methods aligned with diffusion's non-autoregressive principle, we first analyze the intrinsic decoding behavior of dLLMs and their differences from AR models in §4.

# 4 UNDERSTANDING MASK DIFFUSION MODELS BASED ON DIFFUCODER

Current dLLMs such as LLaDA (Nie et al., 2024) and Dream rely on low-confidence remasking decoding strategies (Chang et al., 2022), and LLaDA achieves improved performance on certain tasks using semi-AR decoding methods (*i.e.*, block diffusion decoding; see Arriola et al., 2025 and Fathi et al., 2025). Another common practice among dLLMs is to set the number of diffusion timesteps equal to the sequence length, effectively resorting to token-by-token generation to enhance performance. Given this, we introduce local and global *autoregressive-ness* (AR-ness) metrics to systematically investigate the decoding order of dLLMs. Specifically, our analysis aims to demystify: (1) how dLLMs' decoding patterns differ from those of AR models; (2) how data modality (*e.g.*, code or math) influences model behavior; and (3) how AR-ness evolves across different training stages.

## 4.1 AUTOREGRESSIVE-NESS IN GENERATION

In standard AR decoding, the model generates tokens in strict left-to-right order, ensuring strong sequential coherence. However, diffusion-based decoding may choose to recover `[MASK]` out of order. Therefore, we introduce two metrics to quantify how the unmasking schedule of a diffusion model resembles an AR pattern, including (i) "next token" pattern and (ii) "left first" pattern.

**Local: Consecutive Next-Token Prediction** *Local AR-ness@$k$* is computed by the ratio of predicted sequence matching the pattern of next token prediction within range $k$. If all tokens in $k$-length

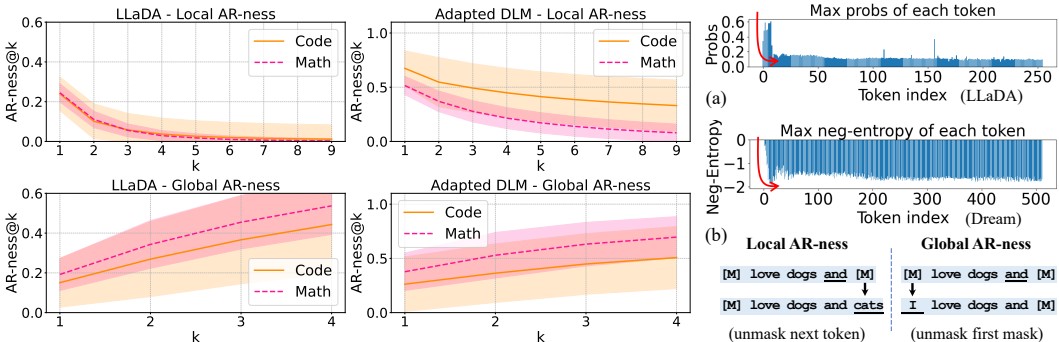

Figure 3: **Left**: Local and global AR-ness across different models and data modalities. Adapted dLLM refers to Dream for the math task and DiffuCoder (Stage 1 trained with 65B tokens) for the code task. **Right**: (a) Confidence score for each position in the dLLM's first forward decoding step (Appx. C.1). (b) Local *AR-ness@k*: the fraction of decoding steps where the newly unmasked token, together with the $k$ immediately preceding predicted tokens, forms a strictly increasing consecutive sequence. Global *AR-ness@k*: the fraction of decoding steps where the model chooses to unmask one of the earliest $k$ positions among all remaining masked tokens.

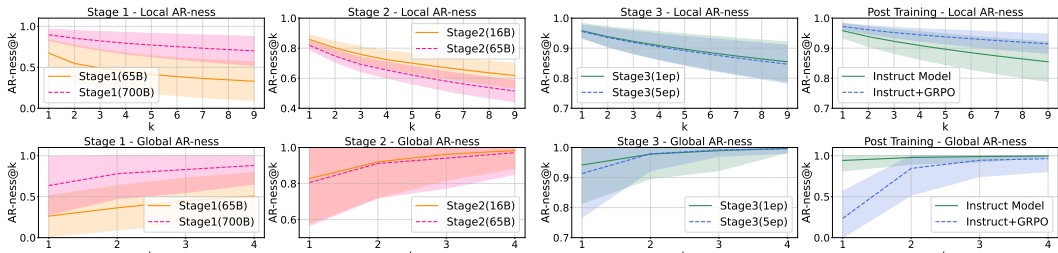

Figure 4: AR-ness drifts on different training stages. **Left**: adaptation pre-training stage and mid-training stage. **Right**: instruction tuning and RL post-training stage.

span are immediate successors of the previously generated token, we count this span as casual. Local AR-ness decays as $k$ grows, as it is harder to maintain longer consecutive spans.

**Global: Earliest Mask Selection**    In step $t$, if the predicted token lies in the first $k$ masked positions, the global AR-ness is scored. *Global AR-ness@k* is the averaged ratio for each $t$, and it measures the tendency to always unmask the earliest remaining token, capturing a left-to-right filling strategy. This ratio grows with $k$, since the criterion becomes easier to satisfy as more early positions are allowed. For the two metrics, the higher value indicates that the generation is more autoregressive. Detailed formulations are listed in Appx. A.2.

## 4.2 DECODING ANALYSIS

We conduct AR-ness comparisons during conditional generation between: (1) different dLLMs, including LLaDA trained from scratch and Dream or DiffuCoder adapted from AR LLMs; (2) different data modalities, including math and code; and (3) different training stages of DiffuCoder. All inference settings are based on the low-confidence remasking strategy (Chang et al., 2022) with the same sequence length and diffusion timesteps (512). Math is evaluated on GSM8K using 8-shots (Cobbe et al., 2021), and code is evaluated using zero-shot HumanEval.

**How do dLLMs decode differently from AR models?**    For AR decoding, both local and global AR-ness are identically equal to 1 (*i.e.*, 100% AR). In contrast, as illustrated in Figure 3, dLLMs do not always decode in a purely AR manner. A significant fraction of tokens in dLLM decoding are recovered from neither the leftmost masked token nor the next token. This observation indicates that dLLMs adopt a more flexible decoding order compared to regular AR models. Nevertheless, both

local and global AR-ness are closer to $1$ than $0$, demonstrating that text data inherently exhibit some AR structure, which diffusion-based LMs, regardless of whether they are trained from scratch or adapted from AR models, naturally capture. For further validation, we examine the confidence score of each recovered token in Figure 3(a) and reveal this phenomenon as the *entropy sink* (Appx. C.1), where the confidence distribution displays a characteristic "L"-shaped pattern. We hypothesize that the entropy sink arises because the intrinsic nature of text biases the model toward tokens that lie immediately to the right of the given prefix: those positions receive stronger positional signals and closer context, leading the model to assign them disproportionately high confidence.

**Entropy sink contributes to AR-ness of dLLMs.** Empirically, adapted dLLMs tend to exhibit stronger AR-ness than those trained from scratch. This is because they inherit the left-to-right token dependencies from the original AR training. Lower AR-ness opens up additional opportunities for parallel generation by breaking this dependency (Appx. C.4). Higher AR-ness can also be beneficial; for example, LLaDA often needs to resort to semi-AR (block-wise decoding; Arriola et al., 2025) generation to achieve higher overall performance. In that setting, the block decoder explicitly reintroduces causal bias into the generation process. In DiffuCoder, we argue that the model can decide how causal it is during generation by itself.

**How do different data modalities affect the decoding paradigm?** According to Figure 3, although math and code decoding exhibit different degrees of local AR-ness, a consistent finding is that code generation has a lower mean and higher variance in global AR-ness. This indicates that when generating code, the model tends to produce later tokens first, leaving some early masked tokens un-recovered until much later (Appx. C.3). The reason might be that mathematical text is essentially sequential and usually requires left-to-right computation, whereas code has an intrinsic structure. Consequently, the model often plans token generation more globally, much like a programmer jumping back and forth through code to refine a code implementation.

**How does AR-ness change at different training stages?** In Figure 4 (Stage 1), after training with 65B tokens, we already observe a relatively low AR-ness. However, when we extend the training to 700B tokens, AR-ness increases while overall performance drops (see the table in Appx. B.1). We suspect that the quality of the pre-training data limits performance. Consequently, we choose the Stage 1 65B model as the starting point for Stage 2. During mid-training (Stage 2) and instruction tuning (Stage 3), on the first epoch of high-quality data, the model learns a high causal bias. As it sees more tokens, however, task performance improves (Appx. B.1), while the measured AR-ness starts to decline. This pattern implies that after the first epoch, dLLMs begin to capture dependencies beyond a pure AR order. After GRPO training, the model's global AR-ness also decreases and meanwhile shows less of a performance drop when decoding in half as many steps (Figure 1(c); Appx. C.4).

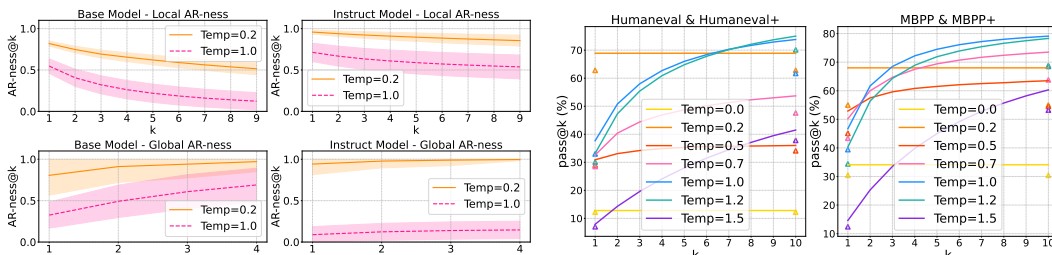

Figure 5: Affects of different sampling temperatures. **Left**: For base model and instruct model, changing temperature affects the AR-ness on HumanEval. **Right**: pass@k curves are different for different temperatures, where triangles refer to score for plus version of each task.

## 4.3 GENERATION DIVERSITY

Post-training studies on AR LLMs (Yue et al., 2025) show that an RL model's reasoning paths are bounded by the base model's pass@k sampling capabilities. Therefore, we examine generation diversity with pass@$k$ accuracy in dLLMs. As Figure 5 (right) and Figure 6 illustrate, for both the base and instruct versions of DiffuCoder, a low temperature yields high pass@1 but little growth in pass@$k$, indicating that the samples lack diversity.

By increasing the temperature to a suitable range (*e.g.*, 1.0 to 1.2), pass@$k$ rises significantly, revealing a latent capability in the model. In many RL settings (Bercovich et al., 2025; Liu et al., 2025a), the model must be able to sample diverse responses during rollouts before RL can reinforce pass@1 accuracy. The promising pass@$k$ curves of DiffuCoder indicate substantial room for improvement through RL, motivating the design of our coupled-GRPO algorithm (§5). Moreover, a higher temperature also substantially lowers AR-ness, as shown in Figure 5 (left) and Figure 1(a), meaning the model generates tokens in a more

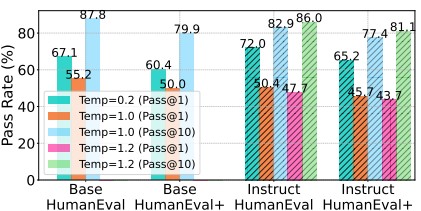

Figure 6: Pass@$k$ scores for different models and temperatures.

random order, which operates differently from AR models. **In AR models, temperature only affects token selection, while in dLLMs it influences both token selection and their generated order.** A visualization of the decoding process from real samples is provided in Appx. C.3.

## 5 COUPLED-GRPO

Reinforcement learning (RL) and GRPO (Shao et al., 2024) have proven critical for enhancing AR models (Bercovich et al., 2025; Shao et al., 2025), but their application to dLLMs is less explored. As discussed in §2.3, formulating the mask diffusion process as a Markov Decision Process allows for a policy optimization approach akin to PPO (Schulman et al., 2017). To facilitate integration with GRPO (Shao et al., 2024), the approximation of token probabilities within diffusion models is necessary. Current masked diffusion models rely on Monte Carlo sampling (Zhu et al., 2025; Shi et al., 2024) for log-probability estimation. Specifically, the negative log-likelihood (NLL) is bounded by the ELBO, *i.e.*, $\mathcal{P}_i = \mathbb{E}_{t \sim 1, \dots T; \boldsymbol{x}_t \sim q(\boldsymbol{x}_t | \boldsymbol{x}_0)}[\mathcal{L}_t(\boldsymbol{x}_t)]$, where $\mathcal{L}_t$ is the cross-entropy loss introduced in Eq. (1). However, Monte Carlo sampling introduces significant overhead during the training of GRPO, as highlighted by d1 (Zhao et al., 2025).

**Baseline methods** To overcome this, d1 chooses to mask all completion tokens and perform a single forward pass to compute each token's probability, which is equivalent to sampling once at diffusion step $t = T$; we call this sequence of log-probabilities $\mathcal{P}_i$, with each element being $\log \pi_\theta(o_i^k | c, o_i^k = \boldsymbol{m})$. Thus, GRPO's update in Eq. (3) uses the ratio $\rho_i^k = \frac{\pi_\theta(o_i^k | c, o_i^k = \boldsymbol{m})}{\pi_{\theta_{\text{old}}}(o_i^k | c, o_i^k = \boldsymbol{m})}$. d1 also randomly masks $p = 15\%$ of the condition tokens to increase sampling diversity, which, in practice, makes the completion-token probability estimates unreliable. In our code experiments, masking condition tokens does not yield a stable reward improvement (Figure 7), probably because code tasks demand higher token-level generation accuracy than math tasks. As a result, we revert to the *completion full mask* version ($p = 0\%$) and use $\mathcal{P}_{t=T}$ as our baseline. Even so, this baseline is biased: as shown in our *entropy sink* analysis (§4.2), high-entropy tokens tend to lie on the left side, so RL training still ends up updating early tokens more aggressively.

**Coupled-GRPO** In the $\mathcal{L}_t$ computation, only the loss for positions involving masked tokens is counted, which introduces inefficiency and variance when sampling times are limited. To improve probability estimation while still covering every token, we introduce a coupled-sampling scheme (Figure 2). Concretely, we pick $\lambda$ timestep pairs $(t, \hat{t})$ with $t + \hat{t} = T$, then sample two complementary completion masks: each mask hides part of the tokens, and together they cover all completion tokens. In other words, every token is unmasked in exactly one of the two forward passes. This design guarantees that (1) each token's log-probability is computed at least once (giving each token a non-zero learning signal) and (2) these log-probability estimations are more accurate because every token is evaluated under a realistic partial-masking context rather than always being masked, and we have $2\lambda$ additional samples compared to the baseline. Combining Eqs. (1), (2), and (3), we have:

$$\mathcal{J}_{\text{GRPO}}(\theta) = \mathbb{E}\Big[\sum_{i=1}^{G}\sum_{k=1}^{|o_i|}\min\Big(\frac{\pi_\theta(o_i^k|c, o_{i,t<T}^k)}{\pi_{\theta_{\text{old}}}(o_i^k|c, o_{i,t<T}^k)}A_i, \text{clip}\Big(\frac{\pi_\theta(o_i^k|c, o_{i,t<T}^k)}{\pi_{\theta_{\text{old}}}(o_i^k|c, o_{i,t<T}^k)}, 1-\varepsilon, 1+\varepsilon\Big)A_i\Big) - \beta D_{\text{KL}}\Big],$$

$$\text{with } \log\pi_\theta(o_i^k|c, o_{i,t<T}^k) = \frac{1}{\lambda+1}\Big[\sum_{t+\hat{t}=T}^{\lambda}[\mathcal{L}_t(\boldsymbol{x}_t) + \mathcal{L}_{\hat{t}}(\boldsymbol{x}_{\hat{t}})] + \mathcal{L}_T(\boldsymbol{x}_T)\Big]_i^k, \ \delta_{\boldsymbol{x}_t,\boldsymbol{m}} + \delta_{\boldsymbol{x}_{\hat{t}},\boldsymbol{m}} = \boldsymbol{1}. \quad (4)$$

In practice, we choose $\lambda = 1$. For a fair comparison, we introduce a de-coupled baseline with the same number of samples but without the complementary constraint. For advantage score computation, we also consider leave-one-out (LOO) strategies (Ahmadian et al., 2024; Kool et al., 2019) to determine the baseline score: $A_i = r(o_i) - \frac{1}{G-1} \sum_{j \neq i}^{G} r(o_j)$, which creates an unbiased estimate. We show that our coupled-sampling scheme can be viewed as an application of the Antithetic Variates (Hammersley & Mauldon, 1956) variance reduction technique in Appx. A.3, where we also list detailed designs for verified rewards, including a code format reward and the execution pass rate over test cases as a correctness reward.

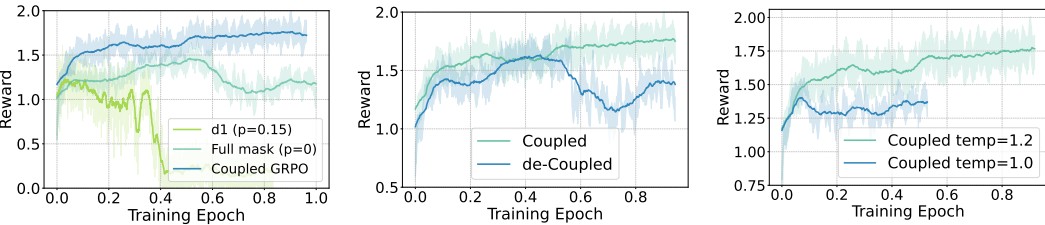

Figure 7: Reward curves during GRPO training. **Left**: Comparison between coupled GRPO and d1 baselines. **Middle**: Decoupled GRPO uses the same number of samplings but with randomly sampled mask noise. **Right**: Coupled-GRPO is sensitive to the rollout temperature.

| Model | HumanEval | | MBPP | | BigCodeBench (C) | | BigCodeBench (I) | |
| --- | --- | --- | --- | --- | --- | --- | --- | --- |
| | – | Plus | – | Plus | Full | Hard | Full | Hard |
| Qwen2.5-Coder+SFT | 82.9 | 75.6 | 80.1 | 66.1 | 46.9 | 16.2 | 39.5 | 14.9 |
| + GRPO | 80.5$_{-2.4}$ | 75.0$_{-0.6}$ | 84.4$_{+4.3}$ | 72.8$_{+6.7}$ | 49.7$_{+2.8}$ | 16.2$_{0.0}$ | 40.0$_{+0.5}$ | 10.2$_{-4.7}$ |
| DiffuCoder-Instruct | 72.0 | 65.2 | 75.1 | 61.9 | 35.7 | 12.2 | 34.0 | 8.8 |
| + coupled GRPO | 73.2$_{+1.2}$ | 68.3$_{+3.1}$ | 78.6$_{+3.5}$ | 67.5$_{+5.6}$ | 40.4$_{+4.7}$ | 10.8$_{-1.4}$ | 37.5$_{+3.5}$ | 10.8$_{+2.0}$ |
| + coupled GRPO (LOO) | 70.7$_{-1.3}$ | 62.2$_{-3.0}$ | 79.6$_{+4.5}$ | 68.5$_{+6.6}$ | 41.2$_{+5.5}$ | 13.5$_{+1.3}$ | 37.6$_{+3.6}$ | 12.8$_{+4.0}$ |
| w/ full mask completion | 66.5$_{-5.5}$ | 59.1$_{-6.1}$ | 77.0$_{+1.9}$ | 65.1$_{+3.2}$ | 38.0$_{+2.3}$ | 8.8$_{-3.4}$ | 35.6$_{+1.6}$ | 13.5$_{+4.7}$ |
| w/ decoupled sampling | 68.9$_{-3.1}$ | 62.8$_{-2.4}$ | 78.3$_{+3.2}$ | 66.4$_{+4.5}$ | 40.4$_{+4.7}$ | 10.8$_{-1.4}$ | 36.5$_{+2.5}$ | 10.8$_{+2.0}$ |

Table 2: Evaluation results for GRPO post-training across multiple benchmarks and models. We report the best results from the sampling temperature set $\{0.2, 0.3, 0.4\}$.

**Experiment Results** Table 2, together with Figure 7, demonstrates the effectiveness of coupled-GRPO training. In contrast, the baseline variants: d1, full-mask completion, and decoupled sampling, exhibit unstable reward learning. The rollout sampling temperature is also critical: as shown in Figure 6, DiffuCoder-Instruct attains a higher pass@10 at temperature 1.2 than at 1.0, mirroring the trend observed during coupled-GRPO training. Notably, RL fine-tuning shifts the optimal sampling temperature from 0.2 to a larger value, such as 0.3 or 0.4, during evaluation, indicating that training sharpens the per-token distribution. This finding aligns with recent results for AR LLMs (Cui et al., 2025; Liu et al., 2025a; Agarwal et al., 2025), suggesting that the approaches proposed in these works may also be generalizable to dLLMs. Finally, at the new optimal temperature, global decoding AR-ness decreases, as shown in Figure 4 (right). A more interesting finding is that, as shown in Figure 1(c), when we use $0.5\times$ fewer decoding steps (equivalent to a $2\times$ generation speedup), training with coupled-GRPO results in a smaller performance drop compared to the model before training, suggesting that AR-ness is reduced and parallelism is increased (Wu et al., 2025). Detailed discussions are provided in Appx. C.4. The comparison for different $\lambda$ is in Appx. C.5

## 6 RELATED WORK

**Text Diffusion Models** Early explorations of text diffusion models were based on a continuous space (Li et al., 2022; Gong et al., 2023b; Chen et al., 2023). Subsequently, discrete diffusion models (Hoogeboom et al., 2021; Austin et al., 2021a) directly introduced discrete noise to accommodate the discrete nature of text, demonstrating significant potential (Zheng et al., 2024; Lou et al., 2024)

and were further developed into mask diffusion models (Shi et al., 2024; Ou et al., 2024; Sahoo et al., 2024). Recent work has explored scaling these models significantly, with DiffuLLaMA (Gong et al., 2025) being adapted from pretrained AR LLMs, and LLaDA (Nie et al., 2024) and Dream (Ye et al., 2025) being the first open-source diffusion LLMs to achieve performance comparable to AR LLMs. Block diffusion (Arriola et al., 2025) proposes a hybrid approach that applies diffusion within each block (Han et al., 2023), serving as a midpoint between autoregressive and diffusion models. Multimodal models such as LaViDa (Li et al., 2025), MMaDA (Yang et al., 2025), and Dimple (Yu et al., 2025) combine text diffusion models with vision models. Liu et al. (2025b); Hu et al. (2025); Ma et al. (2025); Wu et al. (2025); Sahoo et al. (2025) introduce caching and parallel decoding algorithms for dLLMs, significantly improving inference efficiency.

**Code Generation**  Code generation is a crucial domain for LLMs (Roziere et al., 2023; Sun et al., 2024), exemplified by state-of-the-art open-source models like Qwen-2.5-Coder (Hui et al., 2024) and OpenCoder (Huang et al., 2024), with wide applications in areas such as coding assistants and agents (Xu et al., 2024). CodeFusion (Singh et al., 2023) was the first to combine diffusion models with code generation, but it was limited to small-scale models and simple tasks. Recent commercial-scale dLLMs, such as Mercury (Inception Labs et al., 2025) and Gemini (DeepMind, 2025), have demonstrated that diffusion-based code generators can achieve performance comparable to leading autoregressive code models while offering significantly faster generation speeds.

**Reinforcement Learning**  Reinforcement learning with verifiable reward (RLVR) using GRPO (Shao et al., 2024; OpenR1, 2025; Guo et al., 2025; Bercovich et al., 2025) is highly effective in enhancing a language model's math reasoning (Shao et al., 2025) and code generation abilities (Xie et al., 2025). Wang et al. (2025) show the importance of mid-training during RL scaling. Combining RL and diffusion models, VRPO (Zhu et al., 2025) introduces the efficient sampling algorithm from DPO (Rafailov et al., 2023) for dLLMs. d1 (Zhao et al., 2025) and MMaDA (Yang et al., 2025) optimize math reasoning in dLLMs using GRPO, but they rely heavily on block diffusion decoding during rollout and evaluation. LLaDou (Huang et al., 2025) trains an additional module to predict the rank score of tokens. For small text diffusion models, Zhang et al. (2025) propose the target concrete score matching framework, and Zekri & Boullé (2025) introduces score entropy policy optimization (SEPO). Earlier, DDPO (Black et al., 2024) and DPPO (Ren et al., 2025) formulated the diffusion process as a Markov Decision Process and performed policy optimization for continuous diffusion models.

## 7  CONCLUSION

In this work, we present DiffuCoder, a 7B-scale open-source diffusion model for code with strong performance, and its complete training recipe. We also present a comprehensive analysis of dLLMs for code generation. Our investigation into their decoding patterns reveals fundamental differences from AR models; notably, sampling temperature affects not only token selection but also the generation order, creating rich sample diversity for optimization. Capitalizing on this, we introduce coupled-GRPO, a reinforcement learning algorithm that respects the non-autoregressive nature of dLLMs. By using a novel coupled-sampling strategy, our method provides a more accurate likelihood estimation. Coupled-GRPO significantly boosts DiffuCoder's performance, demonstrating the effectiveness of RL methods aligned with diffusion principles. Our work provides the community with a deeper understanding of dLLMs and lays a strong foundation for future explorations of dLLMs in complex reasoning and generation tasks.

## ETHICS STATEMENT

This work develops a diffusion large language model for code generation. Such models could be misused to produce harmful or insecure code. We focus on core methods and train only on public datasets; to our knowledge, they do not include personal or sensitive data. We follow standard filtering and manually review representative generations used in the paper to avoid toxic or misleading content.

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

# A PROBLEM FORMULATION

## A.1 MASK DIFFUSION MODELS

Diffusion models (Ho et al., 2020; Song et al., 2021) contain a forward process that gradually corrupts data $\boldsymbol{x}_0 \sim p_{data}(\boldsymbol{x}_0)$ into noisy variables $\boldsymbol{x}_{1:T}$ through $q(\boldsymbol{x}_{1:T}|\boldsymbol{x}_0) = \prod_{t=1}^{T} q(\boldsymbol{x}_t|\boldsymbol{x}_{t-1})$, and a backward process that models the joint probability as $p_\theta(\boldsymbol{x}_{0:T}) = p_\theta(\boldsymbol{x}_T) \prod_{t=1}^{T} p_\theta(\boldsymbol{x}_{t-1}|\boldsymbol{x}_t)$, denoising $\boldsymbol{x}_t$ to reconstruct $\boldsymbol{x}_0$. The parameters $\theta$ are learned by minimizing the negative log-likelihood via the evidence lower bound (ELBO):

$$-\log p_\theta(\boldsymbol{x}_0) \leq \mathbb{E}_{q(\boldsymbol{x}_1|\boldsymbol{x}_0)}[-\log p_\theta(\boldsymbol{x}_0|\boldsymbol{x}_1)] + D_{\mathrm{KL}}(q(\boldsymbol{x}_T|\boldsymbol{x}_0)||p_\theta(\boldsymbol{x}_T)) + \mathcal{L}_T, \quad (5)$$

$$\text{with } \mathcal{L}_T = \sum_{t=2}^{T} \mathbb{E}_{q(\boldsymbol{x}_t|\boldsymbol{x}_0)}[D_{\mathrm{KL}}(q(\boldsymbol{x}_{t-1}|\boldsymbol{x}_t,\boldsymbol{x}_0)||p_\theta(\boldsymbol{x}_{t-1}|\boldsymbol{x}_t))]. \quad (6)$$

Hoogeboom et al. (2021); Austin et al. (2021a) first proposed discrete diffusion models, where they define the forward process with a categorical distribution $q(\boldsymbol{x}_t|\boldsymbol{x}_{t-1}) = \mathrm{Cat}(\boldsymbol{x}_t; \boldsymbol{Q}_t^\top \boldsymbol{x}_{t-1})$, where $\boldsymbol{x}_t \in \{0,1\}^K$ is a one-hot vector of vocabulary size $K$, and $\boldsymbol{Q}_t \in [0,1]^{K \times K}$ is the transition matrix. A special case is absorbing discrete diffusion, where $\boldsymbol{Q}_t = (1-\beta_t)I + \beta_t \mathbf{1}\boldsymbol{m}^\top$, with $\mathbf{1}$ being an all-ones vector of size $K$ and $\boldsymbol{m}$ the one-hot encoding of a special [MASK] token. Starting from $\boldsymbol{x}_0$, the $t$-step marginal distribution is $q(\boldsymbol{x}_t|\boldsymbol{x}_0) = \mathrm{Cat}(\boldsymbol{x}_t; \boldsymbol{p} = \overline{\boldsymbol{Q}}_t^\top \boldsymbol{x}_0)$, where the cumulative product is $\overline{\boldsymbol{Q}}_t = \prod_{i=1}^{t} \boldsymbol{Q}_i = \alpha_t I + (1-\alpha_t)\mathbf{1}\boldsymbol{m}^\top$, and $\alpha_t = \prod_{i=1}^{t}(1-\beta_t)$. We expect $\alpha_T$ to approach 0 such that the fully noised data $\boldsymbol{x}_T$ equals $\boldsymbol{m}$ with probability 1.

For any two arbitrary time points $0 \leq s < t \leq 1$, the transition distribution between them is $q(\boldsymbol{x}_t|\boldsymbol{x}_s) = \mathrm{Cat}(\boldsymbol{x}_t; \overline{\boldsymbol{Q}}_{s|t}^\top \boldsymbol{x}_s)$, where $\overline{\boldsymbol{Q}}_{s|t} = \overline{\boldsymbol{Q}}_s^{-1}\overline{\boldsymbol{Q}}_t = \frac{\alpha_t}{\alpha_s}I + (1 - \frac{\alpha_t}{\alpha_s})\mathbf{1}\boldsymbol{m}^\top$. This allows us to compute the transition probability between any two timesteps using the ratio of their alpha values:

$$q(\boldsymbol{x}_s|\boldsymbol{x}_t, \boldsymbol{x}_0) = \frac{q(\boldsymbol{x}_t|\boldsymbol{x}_s)q(\boldsymbol{x}_s|\boldsymbol{x}_0)}{q(\boldsymbol{x}_t|\boldsymbol{x}_0)} = \begin{cases} \frac{1 \cdot (1-\alpha_s)}{1-\alpha_t} = \frac{1-\alpha_s}{1-\alpha_t} = 1 - \frac{\alpha_s - \alpha_t}{1-\alpha_t} & \text{if } \boldsymbol{x}_t = \boldsymbol{x}_s = \boldsymbol{m}, \\ \frac{(1-\frac{\alpha_t}{\alpha_s}) \cdot \alpha_s}{1-\alpha_t} = \frac{\alpha_s - \alpha_t}{1-\alpha_t} & \text{if } \boldsymbol{x}_t = \boldsymbol{m} \neq \boldsymbol{x}_s. \end{cases} \quad (7)$$

$$p_\theta(\boldsymbol{x}_s|\boldsymbol{x}_t) = \frac{\alpha_s - \alpha_t}{1-\alpha_t} f_\theta(\boldsymbol{x}_t) + \frac{1-\alpha_s}{1-\alpha_t}\boldsymbol{m}. \quad (8)$$

$$D_{\mathrm{KL}}(q(\boldsymbol{x}_s|\boldsymbol{x}_t,\boldsymbol{x}_0)||p_\theta(\boldsymbol{x}_s||\boldsymbol{x}_t)) = \begin{cases} \frac{\alpha_s - \alpha_t}{1-\alpha_t} D_{\mathrm{KL}}(\boldsymbol{x}_0||f_\theta(\boldsymbol{x}_t)), & \text{for } \boldsymbol{x}_t = \boldsymbol{m}; \\ 0, & \text{for } \boldsymbol{x}_t \neq \boldsymbol{m}. \end{cases} \quad (9)$$

Thus,

$$\mathcal{L}_T = \sum_{t=2}^{T} [-\frac{\alpha_s - \alpha_t}{(t-s)(1-\alpha_t)} \delta_{\boldsymbol{x}_t, \boldsymbol{m}} \boldsymbol{x}_0^\top \log f_\theta(\boldsymbol{x}_t) \Delta_t]. \quad (10)$$

where $\delta_{\boldsymbol{x}_t, \boldsymbol{m}}$ is the indicator function, and $f_\theta(\boldsymbol{x}_t)$ represents the logits of the tokens. In the continuous time limit where $T \to \infty$, we set a small timestep $\Delta_t = t - s = \frac{1}{T} \in (0,1)$. The sum over timesteps becomes an integral, and we have $\alpha_t' = \frac{\alpha_t - \alpha_s}{t - s}$. Following the noise schedule $\alpha_t = 1 - t$, which is widely adopted by Shi et al. (2024); Sahoo et al. (2024); Lou et al. (2024); Ou et al. (2024), we get $\frac{-\alpha_t'}{1-\alpha_t} = \frac{1}{t}$. This can be substituted into Eq. (6), yielding the final ELBO at a sampled time $t$ as a weighted cross-entropy loss:

$$\mathcal{L}_t = \frac{1}{t} \mathbb{E}_{q(\boldsymbol{x}_t|\boldsymbol{x}_0)} \left[ -\sum_{n=1}^{N} \delta_{\boldsymbol{x}_t^n, \boldsymbol{m}} (\boldsymbol{x}_0^n)^\top \log f_\theta(\boldsymbol{x}_t)_n \right]. \quad (11)$$

## A.2 GENERATION AUTOREGRESSIVE-NESS

We formally define local and global autoregressiveness (AR-ness) here.

**Problem Setup** Inference for dLLMs often utilizes low-confidence remasking with the number of diffusion timesteps set equal to the sequence length to ensure high performance. In this setting, let the target sequence length be $L$, and at each diffusion decoding iteration $t = 1, \ldots, T$, the set of still-masked positions just before step $t$ is $M_{t-1} \subseteq \{1, 2, \ldots, L\}$. We denote by $p_t \in M_{t-1}$ the single position unmasked at step $t$, thereby producing the full decoding order $\{p_1, \ldots, p_T\}$.

**Local Autoregressive-ness: Contiguous Next-Token Prediction** For any integer $k \geq 1$, define

$$\mathbb{I}_{\text{local}}(t, k) = \begin{cases} 1, & \text{if } \{p_{t-i}\}_{i=1}^k = \{ p_t - i : i = 1, \ldots, k \}, \\ 0, & \text{otherwise.} \end{cases}$$

The *Local AR-ness@k* is then computed as $\text{NTP@}k = \frac{1}{T} \sum_{t=1}^T \mathbb{I}_{\text{local}}(t, k)$. Local AR-ness measures the tendency to decode the immediate successor of a previously generated token, capturing sequential continuity. It is non-increasing with $k$, as it becomes harder to maintain longer consecutive spans.

**Global Autoregressive-ness: Earliest-First Mask Selection** At step $t$, sort the masked positions $m_{t-1}^{(1)} < m_{t-1}^{(2)} < \cdots < m_{t-1}^{(|M_{t-1}|)}$. Then

$$\mathbb{I}_{\text{global}}(t, k) = \begin{cases} 1, & \text{if } p_t \in \{m_{t-1}^{(1)}, \ldots, m_{t-1}^{(k)}\}, \\ 0, & \text{otherwise.} \end{cases}$$

The *Global FMS-ratio@k* is $\text{FMS@}k = \frac{1}{T} \sum_{t=1}^T \mathbb{I}_{\text{global}}(t, k)$. Global AR-ness measures the tendency to always unmask the earliest remaining token, capturing a front-to-back filling strategy. Together, these ratios reveal how the model behaves during generation. The ratio is non-decreasing with $k$, as the criterion becomes easier to satisfy when more early positions are allowed.

## A.3 COUPLED GRPO

In this section, we provide a detailed formulation of our Coupled GRPO algorithm. As discussed in §5, we improve upon the baseline GRPO by introducing a coupled sampling scheme for more accurate probability estimation. The complete Coupled GRPO algorithm is presented in Algorithm 1.

**Probability Estimation with Coupled Sampling** For a given completion sequence $o$ of length $L$, we select $\lambda$ timestep pairs $(t, \hat{t})$ where $t + \hat{t} = T$. For each pair, we create two complementary masks $M_t$ and $M_{\hat{t}}$ defined as binary vectors in $\{0, 1\}^L$, such that:

$$M_t \vee M_{\hat{t}} = \mathbf{1}, \quad M_t \wedge M_{\hat{t}} = \mathbf{0}, \tag{12}$$

where $\vee$ and $\wedge$ denote element-wise OR and AND, respectively, and $\mathbf{1}, \mathbf{0} \in \{0, 1\}^L$ are the all-ones and all-zeros vectors. The probability estimation for token $o^k$ (also marked as $\boldsymbol{x}_0$) is then computed as:

$$\pi_\theta(o^k | c, o_{t<T}^k) = \frac{1}{\lambda + 1} \left[ \sum_{t+\hat{t}=T}^{\lambda} [\mathcal{L}_t(\boldsymbol{x}_t) + \mathcal{L}_{\hat{t}}(\boldsymbol{x}_{\hat{t}})] + \mathcal{L}_T(\boldsymbol{x}_T) \right]. \tag{13}$$

$\mathcal{L}_t$ is the loss term from Eq (11) at timestep $t$. In detail, we have $\mathcal{L}_t(\boldsymbol{x}_t) = M_t \cdot \frac{1}{t} \cdot \text{CE}(\boldsymbol{x}_t, \boldsymbol{x}_0)$ where CE stands for the cross entropy loss.

**Analysis** The coupled sampling scheme provides several benefits: (i) **Full Coverage**: Each token is guaranteed to be evaluated exactly once in each coupled pair, ensuring complete coverage of the sequence. (ii) **Reduced Variance**: By evaluating each token under realistic partial-masking contexts, we reduce the variance in probability estimates compared to full masking. (iii) **Computational Efficiency**: The coupled sampling requires only two additional forward passes per update compared to the d1 (Zhao et al., 2025) baseline when $\lambda = 1$. The variance reduction can be formally quantified in the next section, Appx. A.4.

## A.4 THEORETICAL ANALYSIS OF COUPLED-GRPO

In this section, we provide a formal analysis of the coupled sampling scheme used to estimate the per-token log-probability proxies within our coupled-GRPO framework. GRPO requires stable estimates of these per-token quantities to compute the importance sampling ratios for the policy gradient update. We demonstrate that our coupled approach can be viewed as a direct and powerful application of the Antithetic Variates (Hammersley & Morton, 1956; Hammersley & Mauldon, 1956) variance reduction technique. We prove that it provides an unbiased estimator for the desired per-token quantity and, critically, that it is guaranteed to reduce estimation variance.

---

**Algorithm 1** Coupled GRPO: Policy Optimization with Coupled Sampling

---

1: **Input:** Reference model $\pi_{ref}$, condition set $\mathcal{C}$, number of completions per condition $G$, code test cases $\mathcal{T}$, hyperparameters $\mu, \beta, \varepsilon$ and $\lambda = 1$
2: Initialize $\pi_\theta \leftarrow \pi_{ref}$
3: **while** not converged **do**
4:     update reference model $\pi_{ref} \leftarrow \pi_\theta$
5:     **for** step $= 1, \ldots, I$ **do**
6:         $\pi_{old} \leftarrow \pi_\theta$
7:         Sample a batch of condition $\mathcal{C}_b \sim \mathcal{C}$
8:         Sample $G$ completions $\{o_i\}_{i=1}^G \sim \pi_{old}(\cdot|c)$, for each $c \in \mathcal{C}_b$
9:         For each $o_i$, compute reward $r(o_i)$ by execute test cases $\mathcal{T}_c$ of each $c$
10:        Get advantage $A_i = r(o_i) - \frac{1}{G}\sum_{j=1}^G r(o_j)$ or LOO $A_i = r(o_i) - \frac{1}{G-1}\sum_{j\neq i}^G r(o_j)$
11:        **for** GRPO iteration $j = 1, \ldots, \mu$ **do**
12:            Randomly sample a timestep pair $(t_j, \hat{t}_j)$ where $t_j + \hat{t}_j = T$
13:            Create complementary masks $M_{t_j}$ and $M_{\hat{t}_j}$ for batch
14:            Compute $\mathcal{L}_{t_j}, \mathcal{L}_{\hat{t}_j}$ and $\mathcal{L}_T$
15:            Compute coupled probability estimates in Eq (13) and importance ratios $\rho_i^k$
16:            Update $\pi_\theta$ by gradient descent on $\mathcal{J}_{\text{GRPO}}$ Eq (4)
17:        **end for**
18:    **end for**
19: **end while**
20: **return** $\pi_\theta$

---

The core challenge is to obtain a stable estimate of a score for each token in a generated sequence, where this score serves as a proxy for its log-probability. This score is defined as an expectation over a random process involving a diffusion timestep $t$ and a mask $M$.

Assuming the linear noise schedule, the sampling process is as follows: A timestep $t$ is drawn from a distribution on $[0, 1]$, typically $t \sim U(0, 1)$. Different from Appx. A.3 which formulates the loss computation batch-wise, in this section we examine the token-wise formulation. Conditional on $t$, a mask for each component in the sequence is sampled independently from a Bernoulli distribution $M_k \sim \text{Bernoulli}(t), k = 1, \ldots, L$.

For each token $k$ in a sequence $o$, we define a per-token scoring function, $g(t, M, k)$, which is non-zero only if token $k$ is masked:

$$g(t, M, k) = M_k \cdot \frac{1}{t} \cdot \ell(o^k | c, o_{\mathbf{1}-M}), \tag{14}$$

where $\ell(\cdot)$ is the cross-entropy loss for token $o^k$ given the condition $c$ and the unmasked context $o_{\mathbf{1}-M}$. The quantity we wish to estimate for each token $k$ is its expected score:

$$v_k = \mathbb{E}_{t,M}[g(t, M, k)]. \tag{15}$$

This estimated $v_k$ is then used to compute the policy probability ratio $\pi_\theta/\pi_{\text{old}}$ in the GRPO objective.

### A.4.1 STANDARD VS. COUPLED MONTE CARLO ESTIMATORS

**Standard MC Estimator.** To estimate the vector of scores $(v_1, \ldots, v_L)$, one can draw $2N$ i.i.d. pairs $\{(t_i, M_i)\}_{i=1}^{2N}$. The estimator for each token $k$ is:

$$\hat{v}_{k,\text{MC}} = \frac{1}{2N} \sum_{i=1}^{2N} g(t_i, M_i, k). \tag{16}$$

In any given sample $i$, $g(t_i, M_i, k)$ is non-zero only for the subset of tokens $k$ where $M_{i,k} = 1$. Many samples are needed to obtain a reliable, non-zero estimate for all tokens.

**Coupled (Antithetic) Estimator.** Our coupled sampling method generates antithetic pairs. We draw $N$ pairs $\{(t_i, M_i)\}_{i=1}^N$ and deterministically create their counterparts $(\hat{t}_i, \hat{M}_i) = (1 - t_i, \mathbf{1} - M_i)$.

The antithetic variates (AV) estimator for $v_k$ is:

$$\hat{v}_{k,\text{AV}} = \frac{1}{N} \sum_{i=1}^{N} \frac{g(t_i, M_i, k) + g(\hat{t}_i, \hat{M}_i, k)}{2}.$$

(17)

A key structural property emerges here: for any given token $k$ and sample $i$, exactly one of the two terms in the inner sum is non-zero. This is because $M_{i,k}$ and $\hat{M}_{i,k} = 1 - M_{i,k}$ are binary complements. This guarantees that every token receives a non-zero score contribution from every coupled pair, ensuring full coverage and making the estimation process vastly more efficient.

### A.4.2 PROOFS

**Unbiasedness.** We show that $\hat{v}_{k,\text{AV}}$ is an unbiased estimator of $v_k$.

*Proof.* The proof relies on showing that the joint probability distribution of $(t, M)$ is identical to that of its antithetic counterpart $(\hat{t}, \hat{M})$. As proven in the previous version, under the symmetric sampling scheme for $t$ and the dependent Bernoulli sampling for $M$, we have $p(t, M) = p(\hat{t}, \hat{M})$.

Since the random variables $(t, M)$ and $(\hat{t}, \hat{M})$ are identically distributed, the expectation of any function of these variables is the same:

$$\mathbb{E}[g(t, M, k)] = \mathbb{E}[g(\hat{t}, \hat{M}, k)] = v_k.$$

(18)

By linearity of expectation, the expectation of the AV estimator is:

$$\mathbb{E}[\hat{v}_{k,\text{AV}}] = \frac{1}{2N} \sum_{i=1}^{N} (\mathbb{E}[g(t_i, M_i, k)] + \mathbb{E}[g(\hat{t}_i, \hat{M}_i, k)]) = \frac{1}{2N} \sum_{i=1}^{N} (v_k + v_k) = v_k.$$

(19)

Thus, the coupled estimator is unbiased. □

**Variance Reduction.** We now provide a direct and rigorous proof that the coupled estimator has lower variance than the standard MC estimator.

*Proof.* The variance of the AV estimator for token $k$ is given by:

$$\text{Var}(\hat{v}_{k,\text{AV}}) = \frac{\text{Var}(g(t, M, k)) + \text{Cov}(g(t, M, k), g(\hat{t}, \hat{M}, k))}{2N}.$$

(20)

Variance is reduced if and only if the covariance term is negative. Let us analyze the covariance:

$$\text{Cov}(g(t, M, k), g(\hat{t}, \hat{M}, k)) = \mathbb{E}[g(t, M, k) \cdot g(\hat{t}, \hat{M}, k)] - \mathbb{E}[g(t, M, k)] \cdot \mathbb{E}[g(\hat{t}, \hat{M}, k)].$$

(21)

Consider the product term inside the first expectation:

$$g(t, M, k) \cdot g(\hat{t}, \hat{M}, k) = \left( M_k \cdot \frac{1}{t} \ell(\dots) \right) \cdot \left( \hat{M}_k \cdot \frac{1}{\hat{t}} \ell(\dots) \right).$$

(22)

The crucial insight is that the product of the mask indicators $M_k \cdot \hat{M}_k$ is always zero, since $\hat{M}_k = 1 - M_k$ and $M_k$ is either 0 or 1. Therefore, the product of the scoring functions is identically zero for all possible values of $t$ and $M$. This means its expectation is also zero:

$$\mathbb{E}[g(t, M, k) \cdot g(\hat{t}, \hat{M}, k)] = 0.$$

(23)

Substituting this back into the covariance formula:

$$\text{Cov}(g(t, M, k), g(\hat{t}, \hat{M}, k)) = 0 - (\mathbb{E}[g(t, M, k)]) \cdot (\mathbb{E}[g(\hat{t}, \hat{M}, k)])$$

(24)

$$= -v_k \cdot v_k$$

(25)

$$= -v_k^2.$$

(26)

Since the loss $\ell(\cdot)$ is non-negative, the scoring function $g$ is non-negative. Its expectation, $v_k$, must therefore be non-negative. Assuming there is some configuration where a loss is incurred (i.e., $v_k > 0$), we have:

$$\text{Cov}(g(t, M, k), g(\hat{t}, \hat{M}, k)) = -v_k^2 < 0.$$

(27)

The covariance is guaranteed to be negative. The variance reduction is therefore not just plausible but a mathematical certainty of this estimation scheme. The amount of reduction is:

$$\text{Var}(\hat{v}_{k,\text{MC}}) - \text{Var}(\hat{v}_{k,\text{AV}}) = \frac{\sigma_g^2}{2N} - \frac{\sigma_g^2 - v_k^2}{2N} = \frac{v_k^2}{2N} > 0. \tag{28}$$

$\square$

This result stems directly from the mutually exclusive nature of the estimators for a given token within a coupled pair, a direct consequence of the complementary masks. It is worth noting that in antithetic sampling, the key requirement is negative correlation between paired samples, not symmetry of the underlying distribution. Pairing $t$ with $T - t$ ensures that the two masks have complementary masking ratios over the sequence. This holds for general monotonic schedules, even if $\alpha_t$ is not symmetric around $T/2$.

### A.5 RELATIONS BETWEEN AR-NESS AND COUPLED SAMPLING

The AR-ness analysis directly influenced the design of Coupled-GRPO in two ways: (1) AR-ness reveals that diffusion models over-update early (left-most) positions. Our measurements show a strong entropy sink at the prefix: the model produces very confident logits there, so standard d1-style sampling causes GRPO to repeatedly update only these early tokens. This leads to biased assignment and slows down learning for later positions. Coupled-GRPO introduces complementary masking to ensure that later tokens are also sampled and updated, directly addressing this imbalance uncovered by AR-ness. (2) AR-ness decreases with higher temperature, enabling more parallel decoding which is important for RL. We observe that higher sampling temperatures reduce AR-ness and produce more diverse, arbitrary-order decoding. Such diversity is essential for RL rollouts. However, this shifts probability mass to later positions, making it even more important to sample and train on those positions reliably. Coupled-GRPO exactly provides the balanced sampling across the entire sequence instead of focusing on the early positions.

Coupled-GRPO not only uses AR-ness insights but also feeds back to produce more parallel, less AR-biased diffusion behavior. As in Appx. C.4, after training, we find that Coupled-GRPO reduces AR-ness and makes the model more robust to step-truncation: using $0.5\times$ fewer diffusion steps ($2\times$ speedup) incurs a smaller accuracy drop compared to the pre-RL model.

## B IMPLEMENTATION DETAILS

### B.1 TRAINING DETAILS

**Adaptation pretraining** During pre-training, we filter the code pre-training corpus from Re-fineCode (Huang et al., 2024) and Stackv2 (Lozhkov et al., 2024). Since RefineCode only provides each item's index in Stackv2, we built a local index engine to download the raw data from Stackv2. We also used text and math data from Fineweb (Penedo et al., 2024) and DCLM-pro (Zhou et al., 2024). The final processed dataset contains around 400B tokens, as shown in Table 3. We adopted the code-to-text ratio suggested in Qwen-2.5-Coder (Hui et al., 2024) and OpenCoder (Huang et al., 2024). The training was conducted on 10 nodes of 8 H100 GPUs each, using `BF16` and full-shard FSDP (Zhao et al., 2023). The total wall-clock time for training on 65B tokens (100,000 global steps) was approximately 40 hours. The single-GPU batch size was 2, and the context window was 4096. Following LLaDA (Nie et al., 2024), we truncated 1% of the data to a random length to improve handling of variable-length inputs. Additionally, for another 1% of the data, we used a random-length prefix as the condition during the diffusion process, which was kept unnoised. We used the Adam optimizer with a maximum learning rate of `2e-5`, with a linear warmup of 20,000 steps followed by a cosine decay schedule to 10% of its peak value at the end of training. The attention mask annealing was performed over 10,000 steps, following DiffuLLaMA (Gong et al., 2025). In our experiments, we observed that training on more tokens in Stage 1 did not improve performance on downstream tasks (Table 4), nor did it lead to further improvements in Stage 2. Therefore, we used the model trained with 65B tokens as our Stage 1 model. This is a well-known behavior in continual pre-training: when a strong base model is trained on high-quality proprietary data, extending training on lower-quality or distribution-mismatched data can hurt downstream performance. In our case, Qwen2.5-Coder is

trained with high-quality, non-public data. Stage 1 adaptation uses an open-source mixture whose quality is inevitably lower, so degradation at long training horizons is expected. With a higher-quality pre-training corpus, we might have reached different conclusions.

| Source | # tokens | Sample weight | Percentage |
|---|---|---|---|
| RefineCode from Stackv2 (code) | 330B | 1 | 78% |
| Fineweb code page (text) | 55B | 1 | 20% |
| DCLM subset (text) | 33B | 1 | |
| Fineweb math page (math) | 3B | 3 | 2% |

Table 3: Adaptation Training Data Recipes on Stage 1

**Mid-training** We used around 16B tokens of annealing code data (Huang et al., 2024) during mid-training. This dataset contains an algorithmic corpus and synthetic data, such as high-quality code snippets and code textbooks. This high-quality mid-training data significantly improves the capacity of the base model, and we chose the model trained on 65B tokens (roughly 4 epochs over the training data) as the final version of DiffuCoder Base. The training was carried out on 8 nodes with 8 A100 GPUs each, using `BF16` and full-shard FSDP (Zhao et al., 2023), which took 90 hours of wall-clock time. We used the Adam optimizer with a maximum learning rate of `1e-5` and a linear warmup of 2,000 steps. Other settings were the same as in Stage 1.

**Instruction tuning** We conducted classifier-free guidance SFT (Gong et al., 2023a; Nie et al., 2024; Ye et al., 2024b) for DiffuCoder using 436K instruction tuning samples from OpenCoder (Huang et al., 2024). The classifier-free guidance approach uses a conditional mask to prevent the diffusion process from adding noise to the condition prefix. We used the same chat template as Qwen2.5-Coder (Hui et al., 2024). We trained a new padding token that was used to pack each sample to a fixed length to control the generation length. We compared different SFT strategies for DiffuCoder instruction tuning, including: (i) using a fixed sequence length of 2048 for each sample with a conditional mask; (ii) using a fixed 2048 length but mixing conditional and unconditional training; and (iii) *flexible padding*, where we padded to the maximum sequence length in the current batch instead of a fixed 2048. Finally, we progressively added a conditional mask for the first epoch, considering that Stages 1 and 2 were trained unconditionally, and then trained the remaining 4 epochs using a fixed sequence length of 2048. The SFT code is based on LLaMA-Factory[5]. The training was performed on 8 nodes of 8 H100 GPUs each, using `BF16` and ZeRO2 (Rajbhandari et al., 2020), taking around 24 hours. We used the Adam optimizer with a maximum learning rate of `1e-5` and a linear warmup ratio of 0.1, followed by a cosine decay schedule to 10% of its peak value at the end of training.

**Coupled GRPO** For RL training, for both our model and the ablation baselines, we filtered 21K hard samples from Acecoder-87k (Zeng et al., 2025) with verifiable test cases and trained for one epoch. We used the pass rate of reference solutions in this dataset to filter for questions with a low average (bottom 20%) and high variance (top 40%) pass rate. These were marked as hard samples, yielding the final 21K samples used for GRPO training. We found that filtering the training samples by difficulty into a proper range is important. We used the online sandbox E2B[6] for code execution and reward verification. We trained the models on a single node with 8 H100 GPUs for a wall-clock time of 40 hours. The default GRPO training parameters were: reference model sync steps 64, number of iterations $\mu = 2$, $\beta = 0.01$, $\varepsilon = 0.5$, learning rate `1e-6`, and maximum completion length 256. The rollout parameters were: diffusion timesteps 256, rollout samples $G = 10$, and sampling temperature 1.2. When sampling coupled $t$, we empirically chose a range of $[0.2, 0.8]$ instead of $[0, 1.0]$ to avoid extreme loss values (Figure 8). Despite training for only one epoch with coupled GRPO, we observed that DiffuCoder-Instruct's performance continued to rise as we added more training steps.

We designed a weighted reward for each completion $o_i$ as $r(o_i) = 2.0 \times r_{\text{code}}(o_i) + 0.5 \times r_{\text{format}}(o_i)$. $r_{\text{code}}(o_i)$ is the pass rate on the test cases, evaluated only if $r_{\text{format}}(o_i) = 1$. $r_{\text{format}}(o_i) =$

---

[5] https://github.com/hiyouga/LLaMA-Factory
[6] https://e2b.dev/

Table 4: Raw results on coding tasks for LLMs and dLLMs in 7/8B scale. * denotes that the results are collocated from public reports instead of evaluating by ourselves. BigCodeBench has completion (C) and instruction (I) version of template during evaluation. +SFT means we conduct the same instruction tuning with DiffuCoder-Instruct.

| Model | HumanEval | | MBPP | | BigCodeBench (C) | | BigCodeBench (I) | |
|---|---|---|---|---|---|---|---|---|
| | - | Plus | - | Plus | Full | Hard | Full | Hard |
| **Base Models** | | | | | | | | |
| Qwen2.5-Coder | 61.6 | 51.8 | 75.9 | 61.4 | 46.1 | 16.2 | 40.2 | 14.2 |
| OpenCoder* | 66.5 | 63.4 | 79.9 | 70.4 | 40.5 | 9.5 | – | – |
| LLaDA | 35.4 | 30.5 | 50.1 | 42.1 | 18.9 | 4.1 | – | – |
| Dream | 56.7 | 50.0 | 68.7 | 57.4 | 23.6 | 4.1 | – | – |
| DiffuCoder (Stage 2 65B) | 67.1 | 60.4 | 74.2 | 60.9 | 40.2 | 12.8 | 1.8 | 0.7 |
| - Stage 1 65B | 39.0 | 31.7 | 48.4 | 38.3 | – | – | – | – |
| - Stage 1 720B | 31.1 | 23.3 | 38.8 | 31.3 | – | – | – | – |
| - Stage 2 16B | 66.5 | 61.0 | 71.9 | 57.1 | 36.7 | 8.8 | – | – |
| **Instruct Models** | | | | | | | | |
| Qwen2.5-Coder-Instruct | 90.2 | 85.4 | 83.9 | 72.0 | 50.7 | 21.6 | 42.2 | 18.2 |
| Qwen2.5-Coder+SFT | 82.9 | 75.6 | 80.1 | 66.1 | 46.9 | 16.2 | 39.5 | 14.9 |
| OpenCoder-Instruct* | 83.5 | 78.7 | 79.1 | 69.0 | 40.3 | 16.9 | – | – |
| LLaDA-Instruct | 35.4 | 31.7 | 31.5 | 28.6 | 16.5 | 2.7 | 14.1 | 1.4 |
| Dream-Instruct | 57.9 | 53.7 | 68.3 | 56.1 | 10.6 | 0.7 | 11.4 | 2.7 |
| Dream+SFT | 56.7 | 50.6 | 71.7 | 58.7 | 27.4 | 6.1 | 25.9 | 5.4 |
| DiffuCoder-Instruct (5 ep) | 72.0 | 65.2 | 75.1 | 61.9 | 35.7 | 12.2 | 34.0 | 8.8 |
| - 1 epoch | 67.1 | 60.4 | 75.7 | 61.9 | 31.1 | 8.1 | 32.0 | 6.1 |
| - 1 ep flexible padding | 65.2 | 58.5 | 71.2 | 59.5 | 35.4 | 11.5 | 32.4 | 10.1 |
| + coupledGRPO (1ep) | 73.2 | 68.3 | 78.6 | 67.5 | 40.4 | 10.8 | 37.5 | 10.8 |
| + coupledGRPO (2ep) | 70.1 | 65.2 | 79.1 | 65.3 | 42.8 | 14.9 | 39.4 | 13.5 |
| **Commercial Models** | | | | | | | | |
| GPT 4o* | 90.2 | – | 82.2 | – | 49.9 | – | – | – |
| Mercury* | 90.0 | – | 77.1 | – | 45.5 | – | – | – |
| Gemini Diffusion* | 89.6 | – | 76.0 | – | 45.4 | – | – | – |

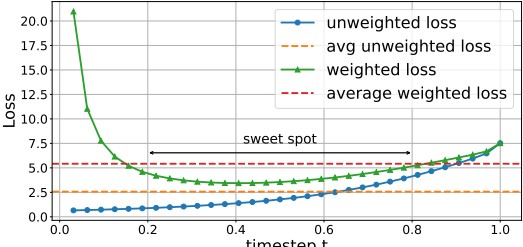

Figure 8: Validation loss distribution of DiffuLLaMA (Gong et al., 2025) for different timesteps. The weighted loss refers to $\mathcal{L}_t$ in Eq. (11), while the unweighted loss refers to the cross-entropy term in the masked diffusion loss without the $1/t$ weighting. Timesteps that are too large or too small will lead to extreme values. The sweet spot is in the span of $[0.2, 0.8]$, which is also consistent for DiffuCoder.

- 0.5 if $o_i$ contains a valid Markdown code block and passes a Python syntax check;

- 0.25 if the code block format is correct but the code has a syntax error;

- 0 if the Markdown block format is not matched.

Concretely, we first require the completion to contain a valid Markdown code block and to pass a Python syntax check (the format reward), before evaluating the test pass rate as the main correctness signal. This gating is important: if code correctness were rewarded without enforcing format, the

model could receive positive rewards while ignoring formatting, leading to progressively degraded structure and harder parsing. In practice, the model quickly saturates the format reward, so the relative upweighting of the test-based component (2.0 vs. 0.5) becomes essential for providing meaningful learning signal.

The raw results are in Table 4. Our design choices in Appx. B.1 rely on these results. DiffuCoder Base underperforms on the instruction-query BigCodeBench (I), likely because the pre-training corpus teaches the base model to focus on completion rather than instruction following. Consequently, it fares better on the subset BigCodeBench (C) with completion query. After instruction tuning, DiffuCoder-Instruct achieves reasonable performance on BigCodeBench (I), indicating that instruction tuning teaches the model to follow instructions.

Table 5: Chat template during the evaluation for HumanEval and BigCodeBench (C).

```
<|im_start|>system
You are a helpful
assistant.<|im_end|>
<|im_start|>user
Please complete the following
problem:
```
{prompt}
```
<|im_end|>
<|im_start|>assistant
Here is the code to solve this
problem:
```python
```

Table 6: Chat template during the evaluation for MBPP and BigCodeBench (I).

```
<|im_start|>system
You are a helpful
assistant.<|im_end|>
<|im_start|>user
```
{prompt}
```
<|im_end|>
<|im_start|>assistant
Here is the code to solve this
problem:
```python
```

## B.2 EVALUATION DETAILS

By default, LLaDA (Nie et al., 2024) employs low confidence remasking with temperature 0, while Dream (Ye et al., 2025) uses top negative entropy remasking with temperature 0.2. Both models use a maximum sequence length and diffusion timesteps of 512. GSM8K tests are conducted using lm-harness[7], and code benchmarks are based on Qwen2.5-Coder's evaluation suitcase[8]. DiffuCoder shares the architecture (Qwen2.5-7B), tokenizer, and inference implementation of Dream-7B. The chat templates we used during the inference are listed below.

## C ADDITIONAL RESULTS

### C.1 ENTROPY SINK

When dLLMs perform conditional generation, the first diffusion step starts with a fully masked completion given a prefix prompt and attempts to recover the completion sequence. At this step, we record the confidence score of each recovered token in Figure 3(a). Appx. C.2 also lists the entropy heatmap across all decoding timesteps. The default decoding algorithm from LLaDA and Dream selects the token with the highest confidence while remasking the rest. LLaDA uses log probabilities while Dream uses negative entropy to measure confidence, where a larger value indicates that the model is highly confident about that token.

Remarkably, the resulting distribution displays a characteristic "L"-shaped pattern. We refer to this phenomenon as the *entropy sink*. We hypothesize that the entropy sink arises because the intrinsic

---

[7] https://github.com/EleutherAI/lm-evaluation-harness
[8] https://github.com/QwenLM/Qwen2.5-Coder/tree/main/qwencoder-eval

nature of text biases the model toward tokens that lie immediately to the right of the given prefix: those positions receive stronger positional signals and closer context, leading the model to assign them disproportionately high confidence. This phenomenon may be related to the cause of the attention sink (Gu et al., 2024; Xiao et al., 2024), but its underlying cause requires further analysis and verification. This entropy bias toward locally adjacent tokens explains why dLLMs still maintain a non-trivial level of AR-ness.

## C.2 ENTROPY PATTERN

In Appx. C.1, we present the phenomenon *entropy sink* and illustrate the L-shaped distribution of the confidence score for a single time step (the first forward). In Figure 9, we extend the visualization to 2D heatmaps. We can still observe a casual bias in these examples, but more token information and flexibility are involved compared to the strict causal mask.

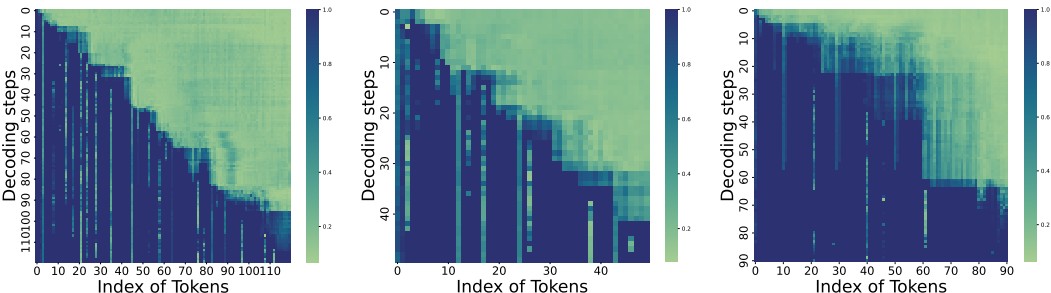

Figure 9: Visualization of the decoding entropy for random samples. The x-axis is the index of generated token, while y-axis refers to decoding steps. Here we set the diffusion timestep and generation length to be equal.

## C.3 DECODING SAMPLES

Figure 10 demonstrates the generation order for different temperatures. The background color runs monotonically from red (earliest) through the spectrum to purple (latest). As we can see, the higher temperature leads to less AR sequence generation. The model tends to determine the right-hand side first, including pad tokens which decide the generation length, and the key parts of this code snippet are generated near the end.

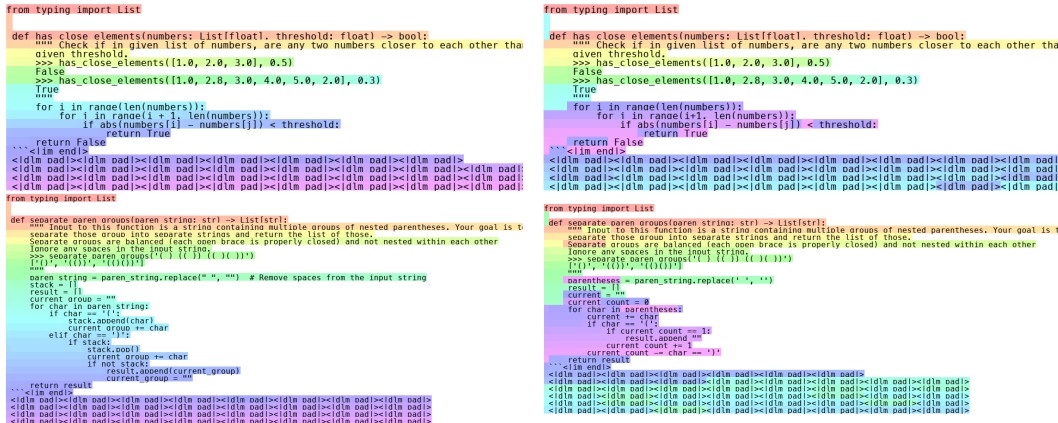

Figure 10: Visualization of the decoding trajectory of DiffuCoder-Instruct under different sampling temperatures. Each character's background is colored from red to purple according to the recover order of the [MASK]. **Left**: temperature is 0.2; **Right**: temperature is 1.2.

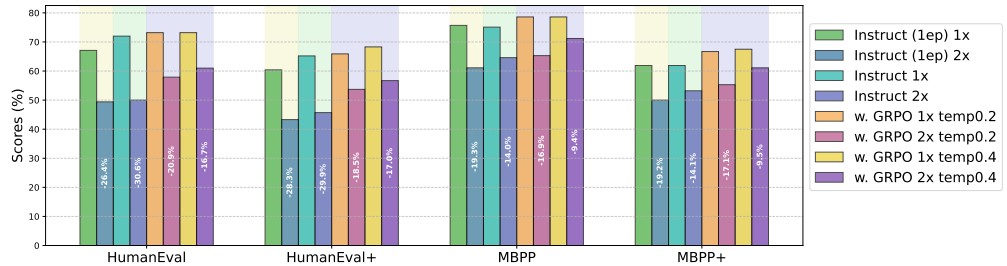

Figure 11: Different model variants act differently when changing decoding timesteps to 1/2 of the sequence length. 1x means the default setting where decoding timesteps are equal to the sequence length while 2x means 1/2 fewer steps which will result in 2x speedup.

## C.4 DECODING TIMESTEPS

Another key motivation for measuring AR-ness is its relationship with generation parallelism. A model with very high AR-ness implies strong left-to-right token dependencies, which limits opportunities for parallel decoding. Conversely, lower AR-ness suggests that the model can generate multiple tokens more independently, enabling fewer diffusion steps and thus faster generation. In other words, by monitoring AR-ness, we also uncover how much headroom remains for accelerated, parallel decoding schedules. In Figure 11, if we correlate the performance drop from 1x to 2x with the non-AR (parallelism) decoding pattern, where a higher drop indicates higher AR-ness, then we can draw the following conclusions. (1) Compared with the Instruct model (the starting point of RL training), GRPO training improves DiffuCoder-Instruct's parallelism. (2) Compared with one epoch of instruction tuning, training for more epochs (5ep here) can reduce the AR-ness of the model. (3) For different sampling temperatures in the GRPO-trained DiffuCoder, the higher temperature (0.4) brings less AR-ness and thus a smaller performance drop at 2x speed.

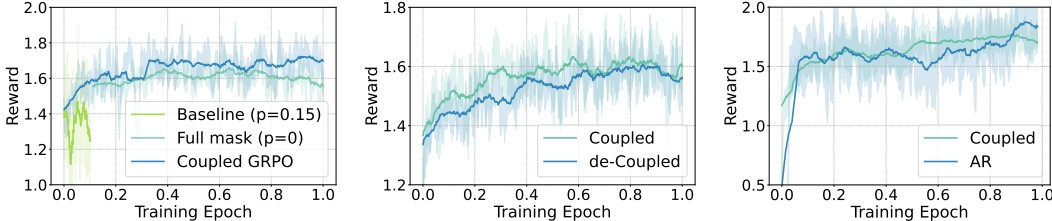

Figure 12: Reward curves during GRPO training. **Left**: Comparison between coupled GRPO and d1 baselines (based on an early version of DiffuCoder-Instruct). **Middle**: Decoupled GRPO uses the same number of sampling times but with randomly sampled masks (based on an early version of DiffuCoder-Instruct). **Right**: Coupled-GRPO on DiffuCoder is compared with regular GRPO for the AR model Qwen2.5Coder+SFT.

## C.5 COUPLED GRPO TRAINING

We monitored the completion length during GRPO training but did not observe a consistent increase in length as seen in AR GRPO training (Shao et al., 2024; OpenR1, 2025). A possible reason is that we do not encourage long-chain reasoning generation (Liu et al., 2025c), which could be a future research direction. In our experimental environment, the end-to-end GRPO training time for DiffuCoder is twice that of the AR model Qwen2.5-Coder. Same with pre-training, post-training of diffusion LLMs is also more compute-intensive than training their AR counterpart due to the sampling and learning efficiency in dLLMs.

We further include two supplementary experiments to clarify the generality of coupled GRPO.

**Effect of larger coupling parameter ($\lambda = 2$).** We add results for $\lambda = 2$ to examine whether increasing the number of complementary mask pairs yields additional benefits. The setting follows

our standard GRPO training on a single node with $8\times$H100 GPUs. As shown in Table 7, $\lambda = 2$ provides only marginal accuracy gains over $\lambda = 1$, while substantially increasing wall-clock time (from 40 to 54 hours). This confirms that $\lambda = 1$ offers the best variance-accuracy-compute trade-off.

Table 7: Performance and cost comparison across $\lambda$.

| $\lambda$ | Theoretical Variance | Training Reward Curve | Wall-clock Time | MBPP | MBPP+ |
|---|---|---|---|---|---|
| 0 (d1) | High | Unstable | 34 hrs | 0.0 | 0.0 |
| 1 (ours) | Low | Stable | 40 hrs | 78.6 | 67.5 |
| 2 | Low | Stable | 54 hrs | 78.8 | 67.2 |

**Applying Coupled-GRPO to Dream 7B.** To demonstrate that coupled GRPO is *model-agnostic* and not tied to DiffuCoder, we apply the same RL recipe to an independent diffusion model Dream-7B-Instruct (Ye et al., 2025). We follow the identical reward design and rollout setting. Table 8 shows that coupled GRPO also improves Dream 7B particularly on MBPP and MBPP+. This supports that our method generalizes beyond a single architecture or training pipeline.

Table 8: Applying coupled-GRPO on Dream-7B-Instruct.

| Model | HE | HE+ | MBPP | MBPP+ |
|---|---|---|---|---|
| Dream-7B-Instruct | 57.9 | 53.7 | 68.3 | 56.1 |
| + coupled-GRPO | 57.9 | 52.3 | **72.5** | **58.2** |

Overall, these additional results reinforce (i) the diminishing returns and increased cost of large $\lambda$, and (ii) the broad applicability of coupled-GRPO across diffusion LLMs. Although our experiments are on code, coupled-GRPO is also domain-agnostic. The method can be applied directly to any diffusion LLM, including text and multimodal settings, with curated domain-specific RL data.

