# OpenReview forum: "DiffuCoder: Understanding and Improving Masked Diffusion Models for Code Generation"
_ICLR.cc/2026/Conference — ICLR 2026 Poster_

### Official Review · Reviewer_17D4 · 2025-10-19

**Soundness:** 3
**Presentation:** 3
**Contribution:** 3
**Rating:** 6
**Confidence:** 4

**Summary:**

The authors trained a 7B diffusion large language model (dLLM), DiffuCoder, on 130B tokens of code and analyzed its decoding behavior on the “AR-ness” of the its sampling process. The authors’ contributions are two-fold: 1. proposed a global and a local AR-ness metric for dLLMs and used the metrics to analyze the sampling process; 2. proposed coupled-GRPO, a RL technique that adapts GRPO of autoregressive LLMs to dLLMs.

**Strengths:**

1. The proposed AR-ness metrics are intriguing and provide insightful analysis of how and when discrete diffusion models exhibit autoregressive-like behavior. While some obervations may seem intuitive, concluding them in a formal way is a meaningful contribution.
2. The introduction of Coupled-GRPO is a simple yet effective technique for adapting GRPO to discrete diffusion models. Table 1 presents a comprehensive benchmark across various LLMs and dLLMs, showing that DiffuCoder clearly outperforms other diffusion-based counterparts, and that DiffuCoder + Coupled-GRPO further improves upon the base model. I also agree that approaches which make dLLMs overly autoregressive undermine their purpose, so it is encouraging to see these results.

**Weaknesses:**

The two main contributions of the paper are valuable in their own right, but I find it difficult to fully understand how they connect as part of a unified method. While the AR-ness metrics are conceptually interesting and insightful, it remains unclear how they relate to or motivate the proposed Coupled-GRPO. Specifically, how does Coupled-GRPO address or influence the behavior measured by the AR-ness metrics? There is only a brief discussion around line 468 and a few related observations in Appendix D.4, but it would be helpful if the authors could provide a clearer explanation or reference that explicitly connects how Coupled-GRPO contributes to improving non-AR behavior in diffusion models.

For Coupled-GRPO, the method pairs timesteps whose indices sum to T, the total number of diffusion steps. This design implicitly assumes a symmetric noising schedule, where the masking ratios at t and T−t are approximately complementary. It would be helpful for the authors to clarify whether this pairing choice is arbitrary or theoretically justified, and whether they have explored or evaluated alternative coupling strategies under non-symmetric schedules.

**Questions:**

See Weaknesses for major concerns.

Miscellaneous:
The figure legends overlap with the plots, which makes them difficult to read. Please consider reformatting the figures to improve clarity and presentation.

---

> ### Author Response · Authors · 2025-11-24
> **Response to Reviewer 17D4**
>
> > Weakness 1 how AR-ness motivates Coupled-GRPO and cp-GRPO contributes to improving non-AR behavior in diffusion models
>
> We appreciate the reviewer’s thoughtful question. The AR-ness analysis directly influenced the design of Coupled-GRPO in two ways:
> 1. AR-ness reveals that diffusion models over-update early (left-most) positions.
>  Our measurements show a strong entropy sink at the prefix: the model produces very confident logits there, so **standard d1-style sampling causes GRPO to focus on updating only these left tokens.** This leads to biased assignment and slows down learning for later positions. Coupled-GRPO introduces complementary masking to ensure that later tokens are also sampled with proper logits and updated balancedly.
> 2. AR-ness decreases with higher temperature, enabling more parallel decoding which is important for RL.
> We observe that higher sampling temperatures reduce AR-ness and produce more diverse, arbitrary-order decoding. **Such diversity is essential for RL rollouts (mirroring AR-LLMs).** However, this shifts probability mass to later positions, making it even more important to sample and train on those positions reliably. **Coupled-GRPO exactly provides the balanced sampling across the entire sequence instead of focusing on the early positions.**
>
> After training, we find that Coupled-GRPO reduces AR-ness and makes the model more robust to step-truncation: using 0.5x fewer diffusion steps (2x speedup) incurs a smaller accuracy drop compared to the pre-RL model. This suggests that **Coupled-GRPO not only uses AR-ness insights but also feeds back to produce more parallel, less AR-biased diffusion behavior.**
>
> In the current revision, we update these clarifications to Appendix due to the page limit. Later, we will clarify this conceptual linkage in the main paper.
>
> > Weakness 2 whether this pairing choice is arbitrary or theoretically justified, and whether they have explored or evaluated alternative coupling strategies under non-symmetric schedules.
>
> We have detailed theoretically proof about coupled sampling in Appendix B.4. Our construction is a direct application of Antithetic Variates (Hammersley & Morton, 1956), a classic variance-reduction technique, which does not require a symmetric schedule.
> In antithetic sampling, the key requirement is negative correlation between paired samples, not symmetry of the underlying distribution. Pairing $t$ with $T-t$ ensures that the two masks have complementary masking ratios over the sequence, where one covers some positions, the other covers the other positions, resulting in complementary correlation in the Monte-Carlo estimator of token log-probability. This holds for general monotonic schedules, even if (1) noise scheduler $\alpha(t)$ is not symmetric around $T/2$; (2) coupling scheduler is not symmetric.
>
> We did not explore non-symmetric couplings since (1) the current noise scheduler $\alpha(t) = 1 − t$ is a standard choice in mainstream dLLMs and testing on other noise schedules requires the retraining of the whole model; (2) Pairing $t$ with $T-t$ ensures  the full coverage for the whole sequence of tokens, which improves training efficiency.
>
> > Question 1 Figure legends overlap.
>
> Thank you for pointing this out. We will adjust the figure layouts (legend placement, spacing, and font size) to ensure that all plots are clearly readable.

---

> > ### Comment · Reviewer_17D4 · 2025-11-25
> >
> > My concerns are solved and I will raise my score.

---

> > > ### Author Response · Authors · 2025-11-26
> > >
> > > Thank you for the update. We really appreciate your careful reading and reconsideration, and we’re glad the clarifications resolved your concerns.

---

### Official Review · Reviewer_vwfe · 2025-10-30

**Soundness:** 3
**Presentation:** 3
**Contribution:** 2
**Rating:** 6
**Confidence:** 3

**Summary:**

This paper introduces DiffuCoder, a 7B-parameter diffusion LLM (dLLM) for code, and uses it to investigate the decoding behavior of dLLMs. The authors propose "autoregressive-ness" (AR-ness) metrics to quantify how left-to-right dLLM generation is and analyze the effect of sampling temperature on diversity. Based on these insights, they introduce Coupled-GRPO, a reinforcement learning method that uses complementary mask pairs to reduce variance in policy gradient estimation. Experiments show DiffuCoder is competitive with AR baselines and that Coupled-GRPO provides modest performance gains (+4.4% on EvalPlus) and improved decoding parallelism.

**Strengths:**

1- Provides the **first systematic measurement of AR-ness** in diffusion LLMs, with clear metrics and visualizations.

2- **Coupled-GRPO** is theoretically motivated (antithetic variates) and supported by formal variance-reduction proof.

3- Offers a **complete open-source training recipe** for a diffusion-based code LLM, potentially useful for the community.

4- Connects decoding order, sampling temperature, and parallelism, yielding practical insights for efficient generation.

5- Writing and figures are generally clear and reproducible; appendices are thorough.

**Weaknesses:**

### **Incremental Analytical Novelty**
The key observations from the AR-ness analysis (e.g., the effect of temperature, the difference between adapted and from-scratch models) confirm properties that are largely expected from the principles of masked diffusion. The contribution is in the measurement, not the discovery of new phenomena. I would suggest that the authors tone the it down in the contribution sections.

### **Modest and Inconsistent Algorithmic Gains**
The improvements from Coupled-GRPO are relatively small and not uniform across all evaluated benchmarks (Table 2), raising questions about its overall impact.

### **Insufficient Ablations**
The paper lacks critical ablations for its core method, including:
- A sweep of the coupling parameter λ to test if the benefits of "coupled" sampling scale.
- A direct comparison against a "2x decoupled" baseline to isolate the effect of complementary sampling from simply doubling the number of forward passes.
- A clear analysis of the computational cost/overhead of Coupled-GRPO.

### **Missing Baseline**
I understand that **Dream-Coder-7B** is relatively a new baseline, but the absence of a comparison with that model gives an unjust favor to DiffuCoder over other open-sourced dLLMs. I would love to see that comparison added to the tables in the rebuttal period, as the model was released almost two months prior to the submission deadline.

### **Overstated Claims**
Several passages (e.g., on temperature-driven diversity) could be more accurately framed as measurements of known model properties rather than novel discoveries.

### **Missing Citations**
The paper does not fully contextualize its work within the broader literature. Specifically:
- When discussing early masked diffusion formulations (e.g., around line 52), the paper should cite "Your Absorbing Discrete Diffusion Secretly Models the Conditional Distributions of Clean Data" (Ou et al., 2024)[1] as a concurrent and relevant work in this space.
- The discussion of semi-AR decoding and Block Diffusion (Arriola et al., 2025)[2] would be strengthened by also citing the concurrent work "Unifying Autoregressive and Diffusion-Based Sequence Generation" (Fathi et al., 2025)[3], which explores similar hybrid modeling ideas. Properly crediting these concurrent works is important for a complete scholarly narrative.

### **Practical Utility of AR-ness**
Beyond interpretability, what are the actionable insights or design recommendations from the AR-ness analysis? For example, could a model's AR-ness score be used to dynamically adjust the number of diffusion steps during inference for speedup without a major performance drop?

### Minor Typographical or formatting issues
1- Page 4, Section 3: "We observed that training with 700B tokens in Stage 1 led to worse performance than using only 65B tokens" -> This is a key observation. The reasoning is plausible, but could be slightly expanded.

2-Page 5, Table 1 Footnote: "Scores are bolded when our model outperforms LLMs specialized for code (excluding LLaDA or Dream)." -> This is an unusual and potentially confusing bolding criterion. It would be clearer to bold the best score in each column or simply state the comparison in the text.

3- Page 8, Section 5: "w. full mask completion" -> Should be "w/ full mask completion" or "with full mask completion" for formal writing.

### **References**
[1]  Ou et al., "Your Absorbing Discrete Diffusion Secretly Models the Conditional Distributions of Clean Data"

[2] Arriola et al., "Block Diffusion: Interpolating Between Autoregressive and Diffusion Language Models"

[3] Fathi et al., "Unifying Autoregressive and Diffusion-Based Sequence Generation"

**Questions:**

Please refer to **Weaknesses**.

---

> ### Author Response · Authors · 2025-11-24
> **Response to Reviewer vwfe (1/2)**
>
> We thank the reviewer for the positive feedback and constructive suggestions. Below, we provide a point-by-point rebuttal to clarify any possible misunderstandings and revise our draft accordingly.
>
> >Weakness 1 Incremental Analytical Novelty. The contribution is in the measurement, not the discovery of new phenomena.
>
> We appreciate the reviewer’s perspective. Our contribution indeed lies in measuring and characterizing AR-ness in diffusion LLMs, rather than claiming the discovery of an entirely new phenomenon. However, we would like to clarify that, **prior to our submission, there was no systematic, quantitative study of AR-like decoding behavior in diffusion LLMs, especially at the 7B scale and across multiple training stages.**
>
> Existing works such as LLaDA and Dream 7B introduced 6 months ago did not analyze their decoding patterns, temperature effects. We provide (1) **Visualization of decoding order shifts under temperature**, showing that higher temperatures induce significantly more parallel, non-AR generation; (2) **First analysis on a diffusion coder model**, a domain for which no prior decoding-behavior study exists (DreamCoder is contemporaneous with our submission and does not include such analysis). **Our contribution is to provide this missing empirical findings.**
>
> >Weakness 2  Improvements from Coupled-GRPO are relatively small and not uniform across all evaluated benchmarks.
>
> We would like to clarify that the gains are consistent across almost all benchmarks: on 7 out of 8 evaluated subsets we observe significant improvements after Coupled-GRPO. In table2, the only drop appears on BigCodeBench-hard (C), but importantly, the LOO variants do show clear improvements on this subset, indicating that despite the variance, the RL method itself is effective.
>
> Across the four standard code benchmarks (HE/HE+/MBPP/MBPP+), the average absolute gain is 3.5%, with the largest improvement on EvalPlus (+4.4%), which is the more challenging setting. Compared with prior diffusion-RL work (d1 and LLaDA-1.5), which report an average ~2.3% improvement on coding tasks, the gains from Coupled-GRPO are actually stronger, despite being achieved with full diffusion decoding, which is more difficult to train than block-wise decoding (used in d1).
>
> Overall, we believe these results demonstrate that Coupled-GRPO has a meaningful and non-trivial impact.
>
> >Weakness 3 The paper lacks critical ablations and analysis.
>
> **(1) Sweep of the coupling parameter $\lambda$.**
>
> We selected $\lambda$ = 1 as the default because it provides the best balance between log-likelihood estimation accuracy and compute efficiency. We have now added $\lambda \in \\{0,1,2\\}$ results in Appx D.5. As shown, $\lambda$ = 1 achieves stable training and strong accuracy, whereas $\lambda$ = 2 adds substantial compute overhead with only marginal
> additional gains, and $\lambda$ = 0 (no coupling) yields high-variance, unstable training.
>
> | λ (pairs) | Theoretical Variance | Training Reward Curve | Wall-clock Time | MBPP | MBPP+ |
> |-----|-------|-----|--------|------|---|
> | 0 (d1)    | High   | Unstable | 34 hrs   | 0.0   | 0.0    |
> | 1 (ours)  | Low     | Stable   | 40 hrs | 78.6  | 67.5  |
> | 2    | Low   | Stable   | 54 hrs   | 78.8  | 67.2   |
>
> **(2) Comparison against a “2x decoupled” baseline.**
>
> This baseline is already included. In Fig. 7 (middle) and Table 2 (last row), decoupled GRPO uses the same number of forward passes as Coupled-GRPO but with independently sampled masks. It exhibits significantly more unstable reward curves and weaker accuracy improvements, confirming that **the benefit comes from complementary (antithetic) coupling rather than simply doubling sampling.**
>
> **(3) Cost/overhead of Coupled-GRPO.**
>  Please refer to (1).
>
> We have incorporated these results into the revised paper for clarity.
>
> >Weakness 4 Comparison to Dream-Coder-7B.
>
> DiffuCoder and Dream-Coder were developed independently and in parallel, with partially overlapping timelines. Because of this, Dream-Coder results were not available to us during the preparation of the main submission. We fully agree it is a relevant baseline, and we provide a comparison here for completeness.
>
> For the base models, the two systems are comparable across all standard code benchmarks:
> | Model              | HE   | HE+  | MBPP | MBPP+ | BCB-full | BCB-hard | Avg  |
> |---|------|----|------|----|-------|-------|-----|
> | DiffuCoder-7B      | 67.1 | 60.4 | 74.2 | 60.9  | 40.2     | 12.8     | 52.6 |
> | Dream-Coder-7B     | 66.5 | 60.4 | 75.9 | 61.6  | 38.5     | 14.2     | 52.9 |
>
> After RL post-training, the two models again show comparable overall behavior, with each model performing better on different subsets:
>
> | Model                    | HE   | MBPP | BCB-full | BCB-hard |
> |--------------------------|------|------|----------|----------|
> | DiffuCoder-7B + RL       | 73.2 | 78.6 | 37.5     | 10.8     |
> | Dream-Coder-7B + RL      | 82.9 | 79.6 | 37.1     | 17.6     |

---

> > ### Author Response · Authors · 2025-11-24
> > **Response to Reviewer vwfe (2/2)**
> >
> > We note that Dream-Coder applies a different RL dataset construction pipeline (including additional filtering of more recent open-source data), which can influence absolute scores. Overall, the results show that DiffuCoder and Dream-Coder are in the same performance range.
> >
> > We will include a brief version of this comparison in the paper to address concerns about fairness among open-source diffusion LLMs.
> >
> > >Weakness 5 Measurements of known model properties rather than novel discoveries.
> >
> > Same to weakness 1
> >
> > >Weakness 6 Missing Citations.
> >
> > We thank the reviewer for highlighting these related works. In the current submission: Ou et al. (2024) is included in our Related Work when discussing masked diffusion formulations. Due to the page limitation, our related work is currently in Appendix. We agree that Fathi et al. (2025) provides an additional perspective on hybrid AR - diffusion modeling; we added Ou et al. (2024) and Fathi et al. (2025) in the main text of the revised version according to your suggestions.
> >
> > >Weakness 7 Practical Utility of AR-ness.
> >
> > Our primary goal with AR-ness is interpretability, but we agree that it can hint at practical design choices. Empirically, we observe that coupled-GRPO consistently reduces AR-ness, making decoding more parallel. As a result, when we apply a 0.5x reduction in diffusion steps (2x speedup), the post-RL model suffers a noticeably smaller performance drop compared to the pre-RL model. This suggests that **lower AR-ness correlates with greater robustness to aggressive step truncation.**
> > While this indicates a meaningful link between AR-ness and speed–accuracy trade-offs, we do not yet claim a precise functional mapping (e.g., predicting the optimal number of steps from AR-ness). Establishing such a quantitative relationship is an interesting direction for future work, but beyond the scope of this paper.
> >
> > >Weakness 8 Minor Typographical or formatting issues.
> >
> > Thank you for pointing these out. (1) We slightly expand the explanation for the Stage 1 degradation when using 700B tokens, noting its connection to data-quality sensitivity in continual pre-training. (2) We agree that the bolding rule in Table 1 could be confusing; we revise it to a more standard format. (3) We fix the phrasing of “w.’’ to “w/’’.
> >
> > All edits are incorporated in the revised version.

---

> > > ### Comment · Reviewer_vwfe · 2025-11-28
> > >
> > > I appreciate the detailed response. Even though I still have a bit of trouble endorsing the novelty of ARness analysis, I think the paper is now in a much better place. Hence, I will raise my score to an 8.

---

### Official Review · Reviewer_eErP · 2025-10-30

**Soundness:** 4
**Presentation:** 4
**Contribution:** 4
**Rating:** 8
**Confidence:** 5

**Summary:**

This paper presents *DiffuCoder*, a large-scale diffusion language model (DLM) designed for code generation. Unlike traditional DLMs that struggle with fixed decoding order and limited self-correction capability, DiffuCoder introduces a complete four-stage training framework including pretraining, mid-tuning, instruction fine-tuning, and reinforcement learning. A major contribution lies in the proposed coupled-GRPO, a variance-reduced reinforcement learning algorithm that enables effective self-reflective sampling under high-temperature decoding. The authors also conduct detailed behavioral analysis, demonstrating DiffuCoder's flexible, non-autoregressive decoding strategy and strong performance across major code and math benchmarks.

**Strengths:**

* Proposes the first full pipeline for large-scale masked diffusion models in code generation, demonstrating competitive performance with AR models.
* Introduces coupled-GRPO, a principled, theoretically justified improvement over prior RL-based DLM training.
* Deep behavioral analysis offers valuable insight into how diffusion decoding diverges from autoregressive patterns.
* Experiments are comprehensive, including comparisons with prior DLMs, autoregressive baselines, and multiple ablations.
* The paper is clearly written and well-structured, making technical innovations accessible.

**Weaknesses:**

This work is strong across the board — in terms of methodological innovation, clarity of explanation, and thorough ablation studies. I did not find any substantive weaknesses, and I believe the paper easily meets the bar for poster acceptance at ICLR. However, given that the core method could likely generalize to domains beyond code (e.g., open-ended language modeling, multimodal reasoning), it would have been valuable to see some discussion of this potential in the main paper. Including such analysis or preliminary results could elevate the work to oral-level significance.

**Questions:**

1. Have you considered applying coupled-GRPO to existing diffusion backbones like LLaDA or Dream to assess its modularity and transferability?
2. Given that the proposed method enhances performance with only two diffusion blocks, have you observed any performance degradation or instability with longer generation lengths or more complex prompts?
3. Do you expect the entropy-sink decoding behavior to generalize to non-code domains, or is it specific to code and math where token dependencies are more structured?

---

> ### Author Response · Authors · 2025-11-24
> **Response to Reviewer eErP**
>
> We really appreciate the reviewer’s positive feedback and thoughtful comments. Our responses are as follows.
>
> > Weakness 1 Potential of generalization to domains beyond code (e.g., open-ended language modeling, multimodal reasoning).
>
> Thank you for the suggestion. Prior diffusion-LLM work (e.g., LLaDA, Dream) primarily targets general language modeling, and MMaDA focuses on multimodal reasoning; these directions are orthogonal to our goal. We choose to focus on code for three reasons:
> 1. Code generation is a critical foundation for modern agentic systems and is worth standalone study.
> 2. Code provides a clean and verifiable reward signal, making it an ideal testbed to develop and analyze RL for diffusion LLMs.
> 3. Diffusion LMs are naturally suitable for code due to their flexible, non-monotonic decoding order, which aligns with how humans iteratively revise code (section 4.2).
>
> Although our experiments are on code, coupled-GRPO is model-agnostic and domain-agnostic. The method applies directly to any diffusion LLM, including text and multimodal settings. We will clarify this potential in the discussion section.
>
> > Question 1 Have you considered applying coupled-GRPO to existing diffusion backbones like LLaDA or Dream to assess its modularity and transferability?
>
> Thank you for the suggestion. We apply the same RL recipe to an independent diffusion model Dream-7B-Instruct. The following table shows that coupled GRPO also improves Dream 7B on coding benchmarks, particularly on MBPP and MBPP+.
>
> | Model                 | HE   | HE+  | MBPP  | MBPP+ |
> |-----------------------|------|------|-------|-------|
> | Dream-7B-Instruct     | 57.9 | 53.7 | 68.3  | 56.1  |
> | + coupled-GRPO        | 57.9 | 52.3 | **72.5** | **58.2** |
>
>
> > Question 2 Given that the proposed method enhances performance with only two diffusion blocks, have you observed any performance degradation or instability with longer generation lengths or more complex prompts?
>
> We first need to clarify that our method does not rely on block diffusion; we use full diffusion decoding throughout. We assume the reviewer’s reference to “two diffusion blocks” corresponds to the two complementary masks used in our coupled sampling scheme. This pairing affects the RL estimator, not the model architecture.
>
> In terms of sequence length, DiffuCoder performs stably on the benchmarks we evaluate (EvalPlus, BigCodeBench), which involve lengths up to 512–768 tokens. For substantially longer or more complex tasks, e.g., SWE-Bench (10k+ tokens) or problems requiring long-chain CoT reasoning, the model currently struggles. This limitation is primarily due to instruction data coverage and the increased modeling complexity of very long sequences for diffusion LLMs. We view scaling diffusion models to long-context settings and complex reasoning as an important direction for future work.
>
> > Question 3  Do you expect the entropy-sink decoding behavior to generalize to non-code domains, or is it specific to code and math where token dependencies are more structured?
>
> We observe the same L-shaped entropy pattern in both math and code tasks, indicating that **diffusion models tend to trust the left-most (near to prefix) positions regardless of domain**. Math tasks show even stronger AR-ness than code, suggesting that this behavior appears in domains with more causal, narrative structure. Code, being more structured and non-linear, exhibits comparatively weaker AR-ness and can partially “escape” a strict left-to-right bias. Overall, we believe entropy-sink behavior is a general property of diffusion LLMs, while the degree of AR-ness varies by domain.

---

> > ### Comment · Reviewer_eErP · 2025-11-26
> >
> > Thank you for the detailed and thoughtful response. I appreciate the clarification regarding the domain scope and the applicability of coupled-GRPO to other diffusion backbones. The results on Dream-7B-Instruct are particularly helpful in demonstrating transferability. I also acknowledge your explanation around long-sequence limitations and the generality of entropy-sink behavior. These strengthen my overall understanding of the method’s potential and current scope. That said, I still believe the paper is best positioned as a poster at ICLR. While the work is thorough, well-motivated, and technically solid, expanding discussion on generalization beyond code and including a more explicit framing of the method’s limitations in the main text would raise its visibility and impact even further. I maintain my overall score of 8, with a strong endorsement for acceptance.

---

> > > ### Author Response · Authors · 2025-11-26
> > >
> > > Thank you so much for the thoughtful follow-up and the strong endorsement. We're glad that the additional experiments and clarifications helped strengthen the understanding of the method’s generality, and we genuinely appreciate your suggestions on improving the framing and discussion. We will incorporate these improvements in the final version.

---

### Official Review · Reviewer_ZNHR · 2025-11-01

**Soundness:** 3
**Presentation:** 3
**Contribution:** 3
**Rating:** 4
**Confidence:** 4

**Summary:**

This paper is a systematic study and methodological work on applying diffusion language models (dLLMs) to code generation; it introduces AR-ness metrics to quantify decoding order and a diffusion-native RL approach (coupled-GRPO), delivers consistent gains across multiple code benchmarks, and reveals the key phenomenon that temperature affects both token selection and generation order.

**Strengths:**

1.	Proposes local/global AR-ness@k to compare dLLMs vs. AR LLMs across stages and modalities.
2.	Higher temperature makes decoding less AR-like and raises pass@k; real sample trajectories are visualized.
3.	Coupled-GRPO lowers variance via complementary masks and avoids semi-AR bias, yielding stable post-training gains.
4.	From adaptation → mid-training → SFT → RL, with compute setups that aid reproducibility.
5.	When halving steps (2× speed), performance drops less after GRPO.

**Weaknesses:**

1.	Generalization to multi-language, multi-file, or agentic tasks is unclear.
2.	Multi-stage large-scale training plus RL rollouts; dLLM GRPO takes ~2× AR’s wall time.
3.	Even with variance reduction, the training still approximates token log-likelihoods.
4.	Stage-1 at ~700B tokens hurts downstream performance, implying sensitivity to data quality and early stopping.
5.	No λ>1 (multi-pair coupling) or alternative t-distributions; limited analysis of reward weighting and verifiers.

**Questions:**

1.	Please evaluate λ>1 (multi-pair coupling) and alternative t-samplings (e.g., Beta/log-uniform), and report the variance–accuracy–compute trade-offs.
2.	Please add a compute-parity comparison against semi-autoregressive RL (block diffusion / re-masking) with matched rollouts and wall-clock time to quantify the benefit of avoiding AR bias.
3.	Please demonstrate robustness by sweeping syntax/format/test weights, adding static analysis and richer tests, and reporting mean ± σ over 3 seeds.
4.	Please report practicality: end-to-end throughput (tokens/s) and training/inference cost, and comment on whether your method stacks with existing dLLM accelerations.

---

> ### Author Response · Authors · 2025-11-24
> **Response to Reviewer ZNHR (1/3)**
>
> We sincerely thank Reviewer ZNHR for acknowledging our contributions to the DLM research. We wish to address your concerns by providing detailed responses to each of your comments.
>
> > Weakness 1 Generalization to multi-language, multi-file, or agentic tasks is unclear
>
> Thank you for the question. **Our paper focuses on building a coder-oriented diffusion LLM, so our evaluation is centered on code generation quality which is also the core capability required by modern agentic systems.** This focus is consistent with existing practice: many leading models (e.g., Qwen, LLaMA, DeepSeek) train dedicated coder variants separate from their general LLMs [1][2][3]. Our goal in this work is to understand and strengthen masked-diffusion decoding in the coding domain, not to position DiffuCoder as a universal multi-language or agentic model. We agree these are valuable directions, but they are orthogonal to our contributions.
>
> Also, our findings actually suggest strong potential for generalization: diffusion decoding flexibility (e.g., controllable AR-ness, non-monotonic generation) is not specific to Python or code. These properties can naturally benefit multi-file reasoning, tool-use, and agentic workflows that require non-linear editing and cross-span coordination.
>
> > Weakness 2 Multi-stage large-scale training plus RL rollouts; dLLM GRPO takes ~2x AR’s wall time.
>
> For multi-stage large training, it especially includes the mid-training and RL, which is standard in AR LLMs [4] but has been far less explored for diffusion LLMs. Our work establishes this full pipeline for dLLMs, helping close the development gap between diffusion and autoregressive architectures.
>
> We believe the reviewer is raising a general concern that diffusion LLMs require more computation than AR models. This concern aligns with observations in prior work: training of diffusion LLMs is more compute-intensive than training their AR counterpart [5][6]. This is valid both for pre-training and post-training due to the sampling and learning efficiency in dLLMs. However, we want to emphasize that **this is a property of the architecture rather than a limitation of our method.** Furthermore, this additional cost would come with distinct modeling benefits: unlike AR models, dLLMs do not impose a fixed causal order with bidirectional context modeling. They model all positions jointly and can revise earlier generated tokens, which is beneficial for tasks that require global consistency, planning, or non-monotonic editing. We updated these discussions in the appendix of our manuscript.
>
> Overall, the compute overhead is an acknowledged characteristic of current dLLMs, not a flaw specific to our method. We aim to contribute to the diffusion LLM community to make the RL more practical and accurate.
> > Weakness 3 Even with variance reduction, the training still approximates token log-likelihoods.
>
> This concern applies to all diffusion generative models, not specifically to our method. In diffusion models, exact log-likelihood or token-wise log-probabilities are typically intractable, and both training and evaluation rely on Monte Carlo or variational approximations:
> - Continuous and discrete diffusion models are trained via variational bounds / denoising score matching, which optimize a surrogate objective whose gradient coincides with that of the log-likelihood in expectation (e.g., DDPM, score-based models). [7]
> - Likelihood estimates (e.g., via importance sampling or probability-flow ODEs) are also Monte Carlo–based approximations, and are standard practice rather than an exception. [8]
>
> In this sense, **our setting is fully aligned with the established theory of diffusion models: we estimate token log-likelihood contributions via sampling, and with more samples the estimator becomes more accurate.**
> Our contribution is to make this much more practical in the diffusion-LLM RL setting. Coupled-GRPO constructs paired, complementary masks and yields a lower-variance estimator of token log-likelihood differences, while keeping the efficiency. So although the estimator is still approximate (as in all diffusion models), we significantly improve its variance vs. compute trade-off, which is precisely what matters in large-scale RL for dLLMs.
>
> > Weakness 4 Stage-1 at ~700B tokens hurts downstream performance, implying sensitivity to data quality and early stopping.
>
> This is a good observation and also one of our empirical findings. The drop at ~700B tokens is not a limitation of diffusion models but a well-known behavior in **continual pre-training: when a strong base model is trained on high-quality proprietary data, extending training on lower-quality or distribution-mismatched data can hurt downstream performance** [9]. In our case, Qwen2.5-Coder-7B is trained with high-quality, non-public data. Stage-1 adaptation uses an open-source mixture whose quality is inevitably lower, so degradation at long training horizons is expected.

---

> > ### Author Response · Authors · 2025-11-24
> > **Response to Reviewer ZNHR (2/3)**
> >
> > > Weakness 5 No $\lambda$>1 or alternative t-distributions; limited analysis of reward weighting and verifiers.
> >
> > We will reply to it in the Question part.
> >
> > > Q1 Please evaluate $\lambda$>1 (multi-pair coupling) and alternative t-samplings (e.g., Beta/log-uniform), and report the variance–accuracy–compute trade-offs.
> >
> > For multi-pair coupling ($\lambda$ > 1), we ran additional experiments on a single node with 8×H100 GPUs. According to our observation, $\lambda$ = 2 achieves a reward curve and final accuracy very close to $\lambda$ = 1, but with noticeably higher wall-clock time, while $\lambda$ = 0 (no coupling) has much higher variance and fails to reach stable training.
> >
> > | λ (pairs) | Theoretical Variance | Training Reward Curve | Wall-clock Time | MBPP | MBPP+ |
> > |---|---|----|---|--|---|
> > | 0 (d1)    | High  | Unstable  | 34 hrs  | 0.0   | 0.0   |
> > | 1 (ours)  | Low | Stable    | 40 hrs   | 78.6  | 67.5 |
> > | 2    | Low  | Stable  | 54 hrs | 78.8  | 67.2 |
> >
> > For alternative t-samplings, prior discrete diffusion work has explored Beta/log-uniform schedules, but [10–14] have established a standard linear schedule ($\alpha(t) = 1 − t$). Since then, the mainstream dLLMs build upon this unified scheduler. Designing and benchmarking new t-schedules is therefore orthogonal to our main contribution and beyond the scope of this work.
> >
> > > Q2 Please add a compute-parity comparison against semi-autoregressive RL (block diffusion / re-masking) with matched rollouts and wall-clock time to quantify the benefit of avoiding AR bias.
> >
> > We agree this is an interesting direction, but it falls outside the scope of our work. Block diffusion introduces explicit AR bias and KV-cache reuse (for faster decoding), which requires a different decoding regime and substantial engineering changes; and our base model is not block-wisely trained diffusion model.
> > Importantly, our paper does not claim that full diffusion or semi-AR decoding is strictly better. One of our core goals is to demystify decoding behavior. We show that full diffusion models naturally exhibit AR-like biases, which are learned to fit data distribution, while increasing temperature encourages decoding later positions, yielding more flexible generation. This allows the model to decide its preferred decoding order rather than imposing AR structure externally.
> >
> > Prior work (e.g., d1 and Llada-1.5) explicitly avoided full diffusion decoding during RL because it produced low-quality samples and was difficult to train (according to d1); our contribution is to make full diffusion RL stable and effective. **Block diffusion is therefore best viewed as one branch of the dLLM design space, whereas our method targets the more general full-diffusion formulation.** Additionally, although we do not replicate block-RL results directly, we note that our absolute improvements on code tasks (+4.4%) exceed those reported in Llada-1.5 (+2.4%) and d1(+2.3%), which also uses block-style decoding for post-training.
> > A full compute-parity comparison across these two distinct decoding paradigms is valuable future work but beyond the scope of this paper.
> >
> > > Q3 Please demonstrate robustness by sweeping syntax/format/test weights, adding static analysis and richer tests, and reporting mean ± σ over 3 seeds.
> >
> > Appendix B.1 details our reward design, which closely follows the d1[15] setup. We use a gated, stepped reward, where $r_\text{code}$​ (test pass rate) is only evaluated when $r_\text{format}=1$. Concretely, we first ensure the completion has a valid Markdown code block and passes a Python syntax check (full format reward), then use test pass rate as the main learning signal. This design is crucial: if we directly reward code correctness without limit on format, the model might get positive code reward but it is encouraged to ignore formatting, making later’s format less regulated and parsing progressively harder. **In practice, we observed that the model quickly saturates the format reward, so the relative upweighting of test-based reward (2.0 vs 0.5) is what matters**; sweeping the exact numeric values (e.g., 2.0 vs 2.5 vs 3.0) is unlikely to provide additional insight and would be very compute-intensive at our scale.
> > Our current reward already includes syntax checking and unit tests; integrating richer static analysis is orthogonal to our main contribution, so we leave this as future work. Finally, running full sweeps with 3 random seeds for each reward configuration is prohibitively expensive (single run already requires tens of GPU-hours on 8×H100).

---

> > > ### Author Response · Authors · 2025-11-24
> > > **Response to Reviewer ZNHR (3/3)**
> > >
> > > > Q4 Please report practicality: end-to-end throughput (tokens/s) and training/inference cost, and comment on whether your method stacks with existing dLLM accelerations.
> > >
> > > We provide full training cost details in Appendix B.1: Adaptation (80 × H100, 40 hrs), Mid-training (64 × A100, 90 hrs), Instruction tuning (64 × H100, 24 hrs), and GRPO (1 node, 8 × H100, 40 hrs). For inference, DiffuCoder uses the same architecture and decoding implementation as Dream-7B; therefore the end-to-end throughput is unchanged (~20 tokens/s on a single batch). **Our method is fully compatible with existing diffusion-LLM accelerations**; any inference- or sampling-level improvement that applies to Dream applies directly to DiffuCoder without modification.
> > >
> > >
> > > **Reference**
> > >
> > > [1] Qwen2.5-Coder Technical Report https://arxiv.org/abs/2409.12186
> > >
> > > [2] Code Llama: Open Foundation Models for Code https://arxiv.org/abs/2308.12950
> > >
> > > [3] DeepSeek-Coder: When the Large Language Model Meets Programming -- The Rise of Code Intelligence https://arxiv.org/abs/2401.14196
> > >
> > > [4] OctoThinker: Mid-training Incentivizes Reinforcement Learning Scaling https://arxiv.org/abs/2506.20512
> > >
> > > [5] Diffusion Language Models are Super Data Learners https://arxiv.org/abs/2511.03276
> > >
> > > [6] Scaling up Masked Diffusion Models on Text https://arxiv.org/abs/2410.18514
> > >
> > > [7] Variational Diffusion Models https://arxiv.org/abs/2107.00630
> > >
> > > [8] Maximum Likelihood Training of Score-Based Diffusion Models https://arxiv.org/abs/2101.09258
> > >
> > > [9] Don’t Stop Pretraining: Adapt Language Models to Domains and Tasks https://aclanthology.org/2020.acl-main.740/
> > >
> > > [10] Large Language Diffusion Models https://arxiv.org/abs/2502.09992
> > >
> > > [11] Your absorbing discrete diffusion secretly models the conditional distributions of clean data https://arxiv.org/abs/2406.03736
> > >
> > > [12] Simplified and Generalized Masked Diffusion for Discrete Data https://arxiv.org/abs/2406.04329
> > >
> > > [13] Simple and effective masked diffusion language models https://arxiv.org/abs/2406.07524
> > >
> > > [14] Dream 7B: Diffusion Large Language Models https://arxiv.org/abs/2508.15487
> > >
> > > [15] d1: Scaling Reasoning in Diffusion Large Language Models via Reinforcement Learning https://arxiv.org/abs/2504.12216

---

### Author Response · Authors · 2025-11-25
**Overall response**

Dear reviewers:

We deeply appreciate your insightful feedback and valuable suggestions. Based on your reviews, we have made thorough revisions to our manuscript.

**Thank you for Acknowledgment of Our Strengths:**
1. A complete, open-source diffusion LLM training pipeline for code, producing a strong 7B model (eErP, vwfe).
2. A systematic AR-ness analysis that provides clear interpretability of diffusion decoding (eErP, 17D4, ZNHR).
3. Coupled-GRPO, a theoretically grounded variance-reduction method that improves performance and reduces AR bias (eErP, 17D4, ZNHR).

**Our Key Revisions:**
1. Added new experiments: (i) $\lambda>1$ to study scalability beyond a single coupled pair, and (ii) applying coupled-GRPO to Dream-7B to demonstrate model-agnostic generalization ([Appendix D.5](https://openreview.net/pdf?id=58NA3unZj5#page=27.52)).
2. Expanded explanations on why Stage-1 over-training reduces accuracy and clarified the reward design rationale, including the weight choices ([Appendix C.1](https://openreview.net/pdf?id=58NA3unZj5#page=22.38)).
3. Beyond existing empirical findings about AR-ness, we further clarified how AR-ness motivates coupled-GRPO and how coupled-GRPO feeds back to reduce AR-bias and increases decoding parallelism ([Appendix B.5](https://openreview.net/pdf?id=58NA3unZj5#page=22.10)).

Thank you again for your contributions to improving our work. We are happy to address any further concerns or queries.

---

### Meta-Review · Area_Chair_yuNY · 2026-01-07

**Summary:**

This paper presents DiffuCoder, a 7B masked diffusion language model trained on large-scale code data. Reviewers broadly agreed that the work is technically strong. The main contributions include metrics to characterize diffusion decoding order, empirical analysis of temperature-dependent generation behavior, and coupled-GRPO. Initial concerns focused on novelty, computational cost, and sufficiency of ablations.

**Reviewer Concerns:**

Reviewers raised concerns about whether the performance gains from coupled-GRPO are sufficiently large and consistent, the computational overhead of full diffusion RL compared to semi-autoregressive alternatives, and the lack of certain ablations. There were also questions about generalization beyond code. In the rebuttal, the authors added new experiments, including application of coupled-GRPO to an independent backbone (Dream-7B), expanded comparisons to contemporaneous diffusion coders, and clearer discussion of computational cost.

**Reviewer Scores:**

Reviewer scores initially ranged from marginally below threshold to strongly positive. The post-rebuttal trajectory shows an upward shift in favor of acceptance.

---

### Decision · Program_Chairs · 2026-01-26

Accept (Poster)